# Thermo-responsive triple-function nanotransporter for efficient chemo-photothermal therapy of multidrug-resistant bacterial infection

Guangchao Qing[1,2,3,4], Xianxian Zhao[5], Ningqiang Gong[2,4], Jing Chen[2,4], Xianlei Li[2,4], Yaling Gan[2], Yongchao Wang[2,4], Zhen Zhang[2,4], Yuxuan Zhang[2,4], Weisheng Guo[6], Yang Luo [1,7] & Xing-Jie Liang [2,4]

New strategies with high antimicrobial efficacy against multidrug-resistant bacteria are urgently desired. Herein, we describe a smart triple-functional nanostructure, namely TRIDENT (Thermo-Responsive-Inspired Drug-Delivery Nano-Transporter), for reliable bacterial eradication. The robust antibacterial effectiveness is attributed to the integrated fluorescence monitoring and synergistic chemo-photothermal killing. We notice that temperature rises generated by near-infrared irradiation did not only melt the nanotransporter via a phase change mechanism, but also irreversibly damaged bacterial membranes to facilitate imipenem permeation, thus interfering with cell wall biosynthesis and eventually leading to rapid bacterial death. Both in vitro and in vivo evidence demonstrate that even low doses of imipenem-encapsulated TRIDENT could eradicate clinical methicillin-resistant *Staphylococcus aureus*, whereas imipenem alone had limited effect. Due to rapid recovery of infected sites and good biosafety we envision a universal antimicrobial platform to fight against multidrug-resistant or extremely drug-resistant bacteria.

[1] Key Laboratory for Biorheological Science and Technology of Ministry of Education, State and Local Joint Engineering Laboratory for Vascular Implants, College of Bioengineering, Chongqing University, Chongqing 400044, P. R. China. [2] Chinese Academy of Sciences (CAS) Key Laboratory for Biomedical Effects of Nanomaterials and Nanosafety, CAS Center for Excellence in Nanoscience, National Center for Nanoscience and Technology of China, Beijing 100190, P. R. China. [3] Department of Materials and Energy, Southwest University, No. 2 Tiansheng Street, Beibei District, Chongqing 400715, P. R. China. [4] University of Chinese Academy of Sciences, Beijing 100049, P. R. China. [5] Department of Clinical Laboratory Medicine, Southwest Hospital, Army Medical University, Chongqing 400038, P. R. China. [6] Translational Medicine Center, Key Laboratory of Molecular Target & Clinical Pharmacology and the State Key Laboratory of Respiratory Disease, School of Pharmaceutical Sciences & The Second Affiliated Hospital, Guangzhou Medical University, Guangzhou 510260, P.R. China. [7] Nuclear Medicine and Molecular Imaging Key Laboratory of Sichuan Province, Department of Nuclear Medicine, the Affiliated Hospital of Southwest Medical University, Sichuan 646000, P. R. China. Correspondence and requests for materials should be addressed to W.G. (email: tjuguoweisheng@126.com) or to Y.L. (email: luoy@cqu.edu.cn) or to X.-J.L. (email: liangxj@nanoctr.cn)

Sepsis is well-recognized as a potentially life-threatening condition. The case fatality rate may reach up to 30% for sepsis, 50% for severe sepsis, and 80% for septic shock[1]. Sepsis was the most expensive condition treated in United States hospital stays in 2013, at an aggregate cost of $23.6 billion for nearly 1.3 million hospitalizations. Preventing the spread of an existing local infection using various antibiotics plays an essential role in lowering the risk of further deterioration. Unfortunately, prophylactic antibiotic medication inevitably increases the prevalence of multidrug-resistant (MDR) or extremely drug-resistant (XDR) pathogenic bacteria[2,3]. According to statistics from WHO, nearly 80% of the MDR or XDR microorganisms have arisen due to the global overuse or misuse of antibiotics, and infection by these strains is accompanied by severe adverse effects such as thrombophlebitis and epidermal necrolysis[4,5]. Therefore, to achieve the goal of timely infection control, it is essential to develop novel antimicrobial approaches that can treat infections with a minimal antibiotic dose.

Recently, nanotechnology-based antibacterial strategies have been proposed due to their enhanced efficacy and therapeutic index[6,7]. Systems based on smart nanoparticles, which can respond to a variety of exogenous or endogenous stimuli to deliver sustained drug release, are particularly desirable[8,9]. Near-infrared (NIR) light is a favorable exogenous stimulus for such smart systems. Photothermal therapy (PTT) triggered by NIR can penetrate tissues deeply while causing little damage to surrounding areas, and it can also fight against pathogenic cells by disrupting membrane permeability and metabolic signals, denaturing proteins/enzymes, and inducing bacterial death[10–13]. The use of varied inorganic or polymer nanoparticles for bacterial killing based on their inherent photothermal capacity has been intensively reported[14]. Besides, nanoparticles such as AuNPs, iron oxide, graphene, black phosphorus, and other polymer nanoparticles capable of photothermal conversion have been designed to intelligently deliver antibiotics to the infected sites for chem-photothermal therapy under NIR irradiation[15–18]. These synergetic strategies can give full play to the advantages of both and enhance the antibacterial effect, as well as reduce the side effects under the control of external NIR irradiation. IR-780 iodide (IR780) is a NIR fluorescence dye with high and stable fluorescence intensity and have been utilized in photothermal therapy[19]. Different from the inorganic nanomaterials, IR780 has better degradability and lower long-term toxicity, which make it more suitable for in vivo photothermal therapy. Besides, the inherent characteristic of the lipophilicity makes it easy to be wrapped by hydrophobic carriers[10]. Taken together, these properties indicate that NIR light-triggered stimulus-responsive gating materials have potential for application in antibacterial nanosystems.

Compared with other stimulus-responsive nanostructures, thermo-responsive nanostructure (TRN) with a large latent heat of fusion and reversible solid-liquid transition over a narrow temperature range are more suitable for precisely controlled drug release[20–22]. Inspired by the success of TRN in cancer therapy, which can be attributed to their convenient assembly, chemical stability, and high biocompatibility, we envisioned that TRN could be exploited as nanocarriers for effective antimicrobial treatment. In particular, we wanted to exploit the following features of TRN: (1) high biocompatibility and no leakage of encapsulated antibiotics. (2) Convenient incorporation of diverse functional molecules into the nanotransporter. (3) Prompt release of encapsulated antibiotics upon exposure to NIR. (4) Synergistic antibacterial activity involving PTT and antibiotic killing[23,24].

In this proof-of-concept experiment, we develop a smart biocompatible thermo-responsive-inspired drug-delivery nanotransporter (TRIDENT) for synergistic eradication of MDR bacteria through combined chemo-photothermal therapy (Fig. 1).

For this purpose, a TRN formulated from natural fatty acids with a tunable melting point around 43 °C is selected as the hydrophobic vehicle for encapsulation of both imipenem (IMP, a broad-spectrum antibiotic) and IR780 (a photosensitizer molecule). And the phospholipid that made of lecithin and DSPE-PEG2000 is used to wrap the drug-loaded TRN for increasing the biocompatibility of the resultant TRIDENT. The final nanosphere is abbreviated as IMP/IR780@TRN. The robust bactericidal capabilities of TRIDENT against both antibiotic-sensitive and clinical MDR bacteria via chemo-photothermal therapy is validated by in vitro and in vivo experiments. The system is effective at a relatively low dosage of IMP than IMP alone. Taken together, our results suggest that TRIDENT is exceptionally effective against pathogenic bacterial infections and can prevent local infections from developing into sepsis.

## Results

**Synthesis and characterization of TRIDENT.** As illustrated in Fig. 1, we utilized lauric acid (melting point (m.p.): 45 °C) and stearic acid (melting point (m.p.): 69 ~72 °C) to prepare the TRN owing to their outstanding biocompatibility and biodegradability[25,26]. The TRIDENT system was fabricated through the classical nanoprecipitation method with slight modifications[24] (Fig. 2a). The spherical morphology of the TRIDENT with a homogeneous size could be observed in TEM image while a hydrodynamic size about 63.39 ± 7.75 nm with a PDI value of 0.27 ± 0.05 was measured by DLS (Fig. 2b, c). Similar sizes were also observed for the IMP@TRN or IR780@TRN (Supplementary Fig. 1). UV-vis absorbance spectra confirmed the successful coencapsulation of both IMP and IR780 in the TRN. The encapsulation efficiency (EE) and loading efficiency (LE) were 51.62 ± 3.68% and 5.46 ± 0.79% for IMP, and 74.85 ± 6.60% and 7.20 ± 0.86% for IR780 (Fig. 2d). Afterwards, we tested the size and morphology of the TRIDENT in diverse media including water, phosphate buffered solution (PBS), 0.9% NaCl (NS), and Luria–Bertani broth (LB) medium. We observed that TRIDENT kept a relatively consistent hydrodynamic size in the four kinds of solvent for 7 days at 4 °C (Fig. 2e and Supplementary Fig. 2). This suggests that the system is stable, and is therefore suitable for antibacterial applications.

**Thermo-responsive properties of the TRIDENT system.** Firstly, we evaluated the thermo-responsive properties via a series of experiments. Prior to the in vitro evaluation, we chose a low-power-intensity (808 nm, 0.5 W cm$^{-2}$) that was suitable for eliminating undesirable damage to ambient healthy tissues to perform the following experiment[27,28].

Next, we used differential scanning calorimetry (DSC) to evaluate the response of TRN formulated with different weight ratios of LA/SA (LA:SA = 2.5, 3.5, and 4.5). We noticed that the TRN prepared with a LA/SA ratio of 3.5:1 had a melting point around 43 °C, which is suitable for thermo-responsive therapy in a physiological setting[29]. The melting points of IMP/or IR780 loaded TRIDENT showed a slight change (Fig. 3a and Supplementary Fig. 3). In addition, we observed that the temperature increment of the TRIDENT suspension depended on both the IR780 concentration and the duration of NIR exposure (Fig. 3b, c). When the TRIDENT solution at a concentration of 20 μg mL$^{-1}$ (in term of IR780) was exposed to NIR irradiation for 5 min, the temperature increase was ~12 °C, whereas no obvious temperature change was observed in the PBS solution (control). A 12 °C increase could lead to serious damage to bacterial cell wall if the start temperature was 37 °C in vitro and in vivo[30], and, therefore, the NIR-induced IR780-mediated PTT effect can potentially be exploited for direct bacterial killing.

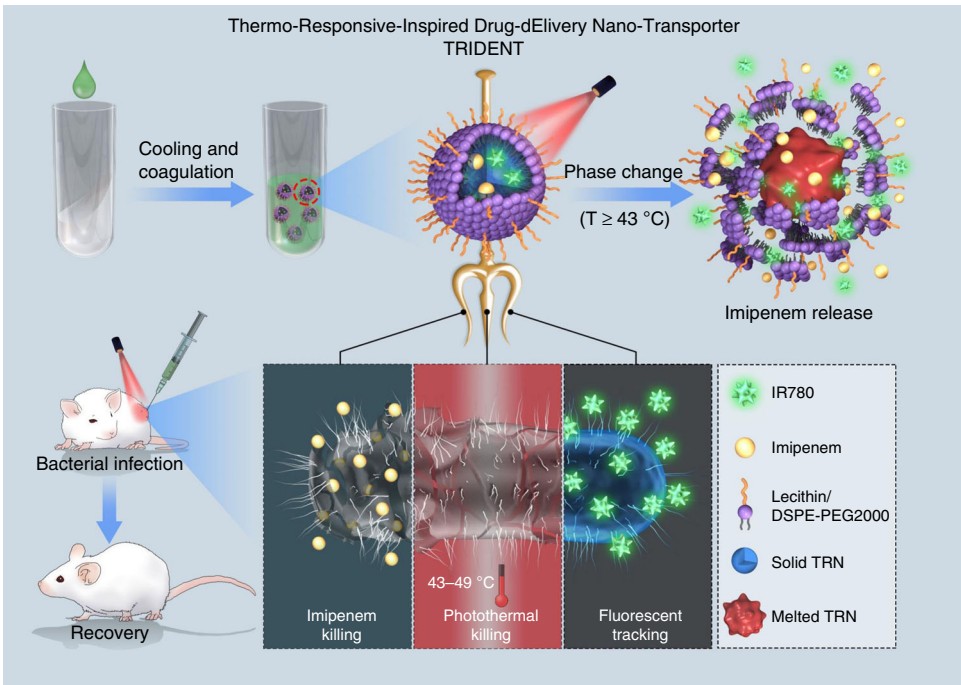

**Fig. 1** Near infrared (NIR)-activated TRIDENT for antibiotic-resistant bacteria killing. The prepared thermo-responsive-inspired drug-delivery nano-transporter (TRIDENT, also named IMP/IR780@TRN) is "melted" when the temperature rises to above 43 °C under the NIR irradiation, leading to the release of imipenem to the infected site. Consequently, the multidrug-resistant bacteria can be efficiently damaged through a synergistic antibiotic-photothermal strategy with a relatively low dosage of antibiotic, eventually accelerating the recovery of the infected skin

We next investigated the morphology of the TRIDENT before and after NIR irradiation. TEM images demonstrated that laser irradiation led to significant change in the particle size but not the morphology of the TRIDENT (Fig. 3d, e and Supplementary Fig. 4). DLS measurements also confirmed this change and the diameters of TRIDENT were decreased to 22.34 ± 1.14 nm after the laser irradiation (insets in Fig. 3d, e and Supplementary Fig. 4). A similar phenomenon was observed for the NIR-irradiated IR780@TRN. And the small particles in TEM image clearly indicated that the IR780@TRNs were damaged after the NIR irradiation (Supplementary Fig. 4).

We also investigated the thermo-responsive antibiotic release characteristics of the nanotransporter under NIR irradiation. There are few IMP was released in the first 3 h, In contrast, after applying the laser irradiation to the TRIDENT solution for 5 min at the point of 3 h, around 15% IMP were rapidly released from the TRIDENT solution within 1 h. More IMP would be released once the illumination time increased. Some IMP would gradually release from the TRIDENT solution within hours at the end of the light and other remained IMP may be rewrapped by the solidified TRN after the temperature of solution dropped to 37 °C (Fig. 3f). These results confirmed that the photothermal effect generated by IR780 upon NIR irradiation could effectively trigger the rapid release of IMP from TRIDENT. A hypothetical model of the morphological change and release of antibiotics under NIR irradiation is illustrated in Fig. 3g.

**In vitro antibacterial activities**. The in vitro cytocompatibility of the TRIDENT was evaluated by testing its toxicity on cell lines derived from normal mammalian tissues including mouse embryo cells (3T3 cells) and human umbilical vein endothelial cells (HUVECs). We observed that the viability of both 3T3 cells and HUVECs were still above 80% even at a TRIDENT concentration up to 80 μg mL$^{-1}$ (Supplementary Fig. 5). This

suggests that the TRIDENT had no obvious cytotoxic effect on normal mammalian cells.

We then investigated the synergistic antibacterial behaviors of the TRIDENT against antibiotic-sensitive *Escherichia coli* (*E. coli*) and *Staphylococcus aureus* (*S. aureus*) under NIR irradiation (Fig. 4). The results from standard plate counting showed that NIR-irradiated TRIDENT (namely IMP/IR780@TRN + NIR) was highly effective against both species and almost no bacterial colony could be found in the plate. The colony of the bacteria treated with IR780@TRN and 3× IMP (three times the dosage of that in TRIDENT) were reduced by 40–50%, but the IMP treatment group based on the theoretical release (T-IMP) did not show obvious antibacterial ability, indicating that photothermal therapy or high doses of IMP alone used in this study did not kill the bacterial cells completely (Fig. 4a). We also explored the thermal response of these bacteria to various temperatures and times relevant to the study to clarify the connection between bacterial viability and temperature. In addition, log$_{10}$ (CFU mL$^{-1}$) (CFU: colony-forming unit) was used to calculate the viability of bacteria treated with different conditions. Around 2log$_{10}$CFU reduction could be monitored when the bacteria were subjected to 49 °C for 5 min. When the temperature was lower than 45 °C, there were no obvious change of the survival rate. The survival rate of bacteria was related to the extent of treatment temperature and time. And the longer the bacteria stayed at one level of temperature, the lower the survival rate would be obtained once the temperature rose to 46°C and above (Supplementary Fig. 6, 7). In the absence of NIR irradiation, the number of colony showed unconspicuous reduction in the group of IMP@TRN, IR780@TRN, and IMP/IR780@TRN, respectively, which was also found in the group of NIR (Fig. 4). After treatment with TRIDENT + NIR, the value of log$_{10}$ (CFU mL$^{-1}$) separately reduced to 2.02 ± 0.22, and 2.04 ± 0.13 for EC and SA, demonstrating that most bacteria were killed compared to control group. While 4.23 ± 0.01 (IR780@TRN + NIR) and 3.73 ± 0.08

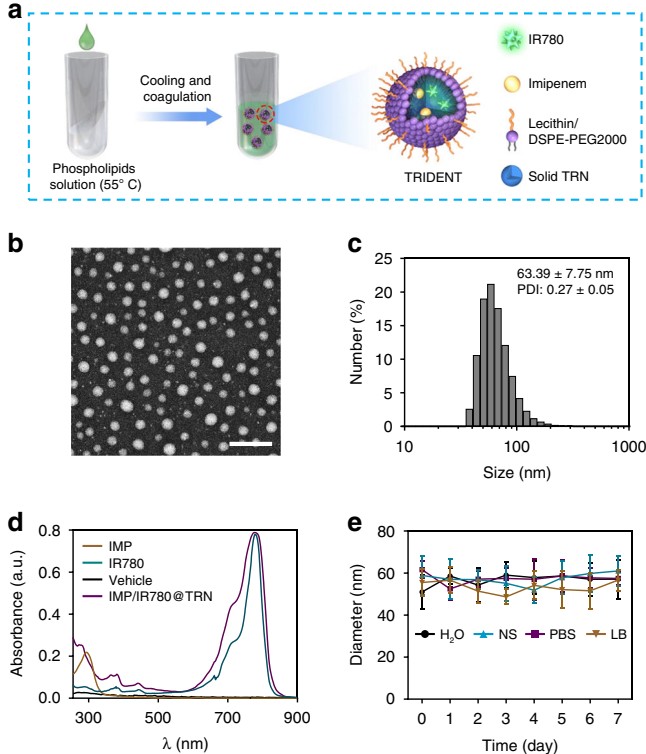

**Fig. 2** Preparation and characterization of TRIDENT. **a** Schematic illustration of the structure of TRIDENT and its construction via modified nanoprecipitation. **b** Representative TEM image of TRIDENT. Scale bar: 200 nm. **c** Particle size distribution histograms of TRIDENT measured by DLS. **d** UV-vis absorption spectra of free IMP, free IR780, vehicle, and TRIDENT. **e** Colloidal stability of TRIDENT in different solutions. NS normal saline, LB lysogeny broth. Data are presented as mean ± s.d. ($n = 4$). Source data are provided as a Source Data file

(3× IMP) for EC and 4.14 ± 0.02 (IR780@TRN + NIR) and 3.83 ± 0.09 (3× IMP) for SA were obtained, no obvious reduction was observed in other treatment groups (Fig. 4b, c). Therefore, we think that the assembled TRIDENT has an extremely strong ability to prevent leakage. Similar results were also observed in the growth-inhibition assay in liquid medium (Supplementary Fig. 8). The bacterial growth was completely inhibited by NIR-irradiated TRIDENT within 48 h, and reduced growth was observed from IR780@TRN + NIR. No noticeable inhibition was observed in the other groups. The bacteria grew rapidly within the first 12 h and then slowed down before reaching a plateau. These results indicate that the TRIDENT, under NIR irradiation, can strongly inhibit the bacterial growth.

The in vitro bactericidal activities were further studied using Live/Dead dual-color fluorescent staining (Fig. 4d). When treated by PBS, IMP@TRN, IR780@TRN, TRIDENT, T-IMP, and NIR alone, most of the bacteria survived (green). In contrast, the proportion of dead bacteria (red) greatly increased in the groups that treated by NIR-irradiated TRIDENT and 3× IMP, which was consistent with the results from the plate count assays and the growth-inhibition assay. Bacterial morphology was also observed by SEM to further evaluate the bactericidal activities of TRIDENT (Fig. 4e, Supplementary Fig. 9). The SEM images showed that NIR-irradiated TRIDENT caused serious damage to the bacterial membrane, and the cell wall fragmentation and irregularly shaped holes could be clearly observed. The killing capabilities of both NIR-irradiated IR780@TRN and 3× IMP were inferior to the NIR-irradiated TRIDENT. In contrast, bacteria treated by other

groups alone had normal shapes and retained clear borders and membrane integrity compared with control group (PBS). We repeated these experiments by replacing the standard antibiotic-sensitive strains with clinically isolated multidrug-resistant *E. coli* (MDREC) and methicillin-resistant *Staphylococcus aureus* (MRSA). Similar results were observed in all groups except MRSA treated with 3× IMP alone, where only partial inhibition of growth was observed (Fig. 5, Supplementary Figs. 8, 9). These results indicate that the synergistic antibacterial behaviors of TRIDENT can be employed to treat those clinical MDR or XDR bacteria that are difficult to deal with using conventional antibiotics.

**In vivo eradication of skin infections**. After establishing that our system has in vitro bactericidal activity, we further evaluated its antibacterial activity in vivo using a mouse model of skin infection with MDREC and MRSA (Fig. 6a). We observed that most TRIDENTs (IMP/IR780@TRN) could effectively stayed in the infected areas after local administration and strong fluorescence was still maintained even up to 48 h, which may be caused by the densification of infected skin tissues[13,31] (Fig. 6b and Supplementary Fig. 10). Thus, the in vivo enrichment of the nanomaterials at the infection site can be monitored in real time, providing visual guidance for the selection of the optimal treatment time. The in vivo photothermal experiments for the developed TRIDENTs were performed afterward (Supplementary Fig. 11). The average temperature of the infected area increased by 12.1 °C at the end of the irradiation. The final temperature could be raised at 49 °C, which would promote photothermal therapy and activate the injected TRIDENT to kill bacteria while reducing the bad influence generated from hyperthermia on surrounding normal tissues. As expected, we found that the mice in the NIR-irradiated TRIDENT group achieved almost full recovery in the two infection models on the 15th day (the observation endpoint) (Fig. 6c–h and Supplementary Fig. 12). And similar results were also observed from 3× IMP group in MDREC-infected mice, but not in MRSA-infected mice that only part of the lesion was recovered. In the IR780@TRN + NIR group, the lesion size was reduced by about 70% and 60% for MDREC-infected and MRSA-infected skin compared to the beginning of the treatment, whereas treated with IMP@TRN, IR780@TRN, IMP/IR780@TRN, T-IMP, and NIR only produced 30–50% reductions. Meanwhile, the body weights of the infected mice in TRIDENT + NIR quickly increased followed by IR780@TRN and 3x IMP, and then by IMP@TRN, IR780@TRN, and IMP/IR780@TRN (Fig. 6e, h). We further validated the bactericidal effect by excising the infected tissues and counting the bacterial CFU on LB petri dishes. The results were identical to the in vitro antibacterial test results shown in Fig. 5. This demonstrates the promising bactericidal potential of NIR-irradiated TRIDENT against MDREC and MRSA infection (Fig. 6i, j).

Next, we performed H&E staining to examine the recovery of the infected skin tissues. Severely disrupted dermis structures and infected skin lesions were observed in the samples from groups treated with PBS, IMP@TRN, IR780@TRN, IMP/IR780@TRN, T-IMP and NIR alone (Fig. 6k and Supplementary Figure 12d). In contrast, the skin tissues obtained from TRIDENT + NIR showed relatively intact histological characteristics, followed by the groups treated with IR780@TRN + NIR and 3× IMP. Together these results reveal the superior therapeutic efficacy of NIR-irradiated TRIDENT. Similar healing processes and therapeutic results were also observed when the infection models were induced with antibiotic-sensitive *E. coli* and *S. aureus* (Supplementary Figs. 13–15). Thus, the proposed strategy had in vivo

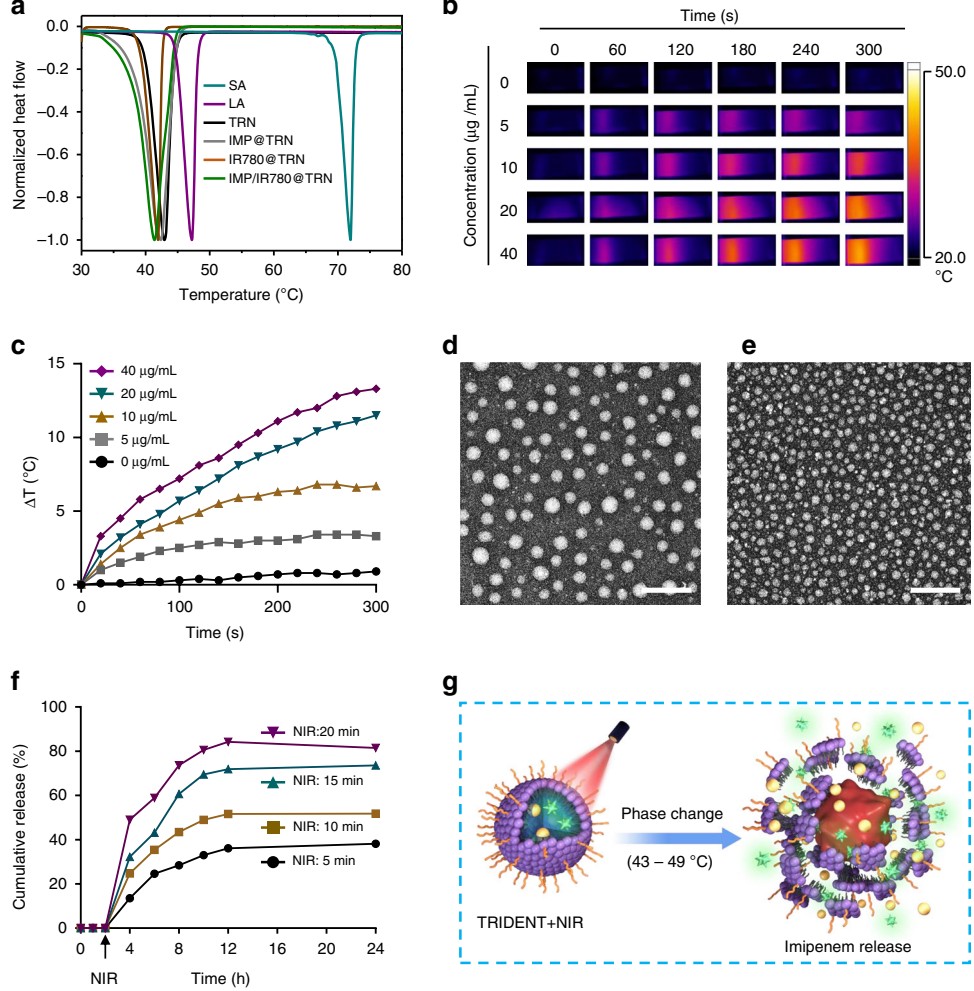

**Fig. 3** Thermo-responsive characteristics of TRIDENT. **a** DSC curves of stearic acid (SA), lauric acid (LA), TRN, IMP@TRN, IR780@TRN and IMP/IR780@TRN. **b** Thermal images of TRIDENT solution under NIR irradiation (808 nm, 0.5 W cm$^{-2}$) at selected time points. **c** Heating curves of TRIDENT solutions under the NIR irradiation. TEM images of TRIDENT, **d** before, and **e** after irradiation for 5 min. Scale bars: 200 nm. **f** In vitro cumulative release profile of IMP from TRIDENT under NIR irradiation with different time. **g** Schematic representation of the thermal phase-change of TRIDENT and the resulting imipenem release. Source data are provided as a Source Data file

antibacterial activities against both antibiotic-sensitive bacteria and clinically relevant MDR bacteria at a low antibiotic dosage.

To better validate the capability of the nanotransporter to curb localized infection, we monitored the change of two major sepsis-associated biochemical markers (procalcitonin (PCT), c-reactive protein (CRP)) in the blood of the MRSA-infected mice after treatment with PBS, NIR-irradiated IMP/IR780@TRN, and 3× IMP (Fig. 7a, b). Normal (non-infected) mice were used as control. As expected, the levels of these two sepsis indicators were greatly increased in the infected mice that treated with PBS compared with control. In the mice treated with NIR-irradiated TRIDENT, the levels of the two biomarkers were similar to those detected in the normal mice. These results demonstrate that the TRIDENT system can prevent the occurrence of severe systemic infection like sepsis from local bacterial infection.

**Biosafety of TRIDENT.** We further performed blood biochemical analysis and histological analysis of mouse samples to evaluate any potential acute biohazards associated with the proposed TRIDENT method. The mice treated with free IMP had increased levels of blood urea nitrogen and creatinine (Supplementary Fig. 16a, b), and hyaline casts were observed in the renal tubules via histological staining (Supplementary Fig. 16c). These data

revealed non-negligible nephrotoxicity was occurred in the mice that treated with 3× IMP[32]. Fortunately, barely changes of the biomarkers were observed the mice treated with PBS and NIR-irradiated TRIDENT. And there were no obvious pathological abnormalities in H&E-stained sections of heart, liver, spleen, lung, and kidney. Overall, these results clearly demonstrated that TRIDENT possesses superior biosafety and can greatly reduce the toxicity of IMP to kidney.

## Discussion

Pathogenic bacterial infections and the corresponding drug resistance have caused serious problems that are undermining the public health[33,34]. Conventional antibacterial strategies pose a substantial risk by triggering increased drug resistance in the post-antibiotic age[35], along with severe adverse effects, which potentially result in toxic symptoms worse than the infection itself[36–38]. To address this challenge, nanoparticle-based targeted delivery strategies have been developed recently to treat bacterial infections with relatively low antibiotic doses and reduced side effects[39–41]. In the field of cancer therapy, significant progress has been made in developing chemically stable, biocompatible and easily prepared thermo-responsive gating materials for precise controlled drug release[42,43]. Inspired by these features, we constructed an

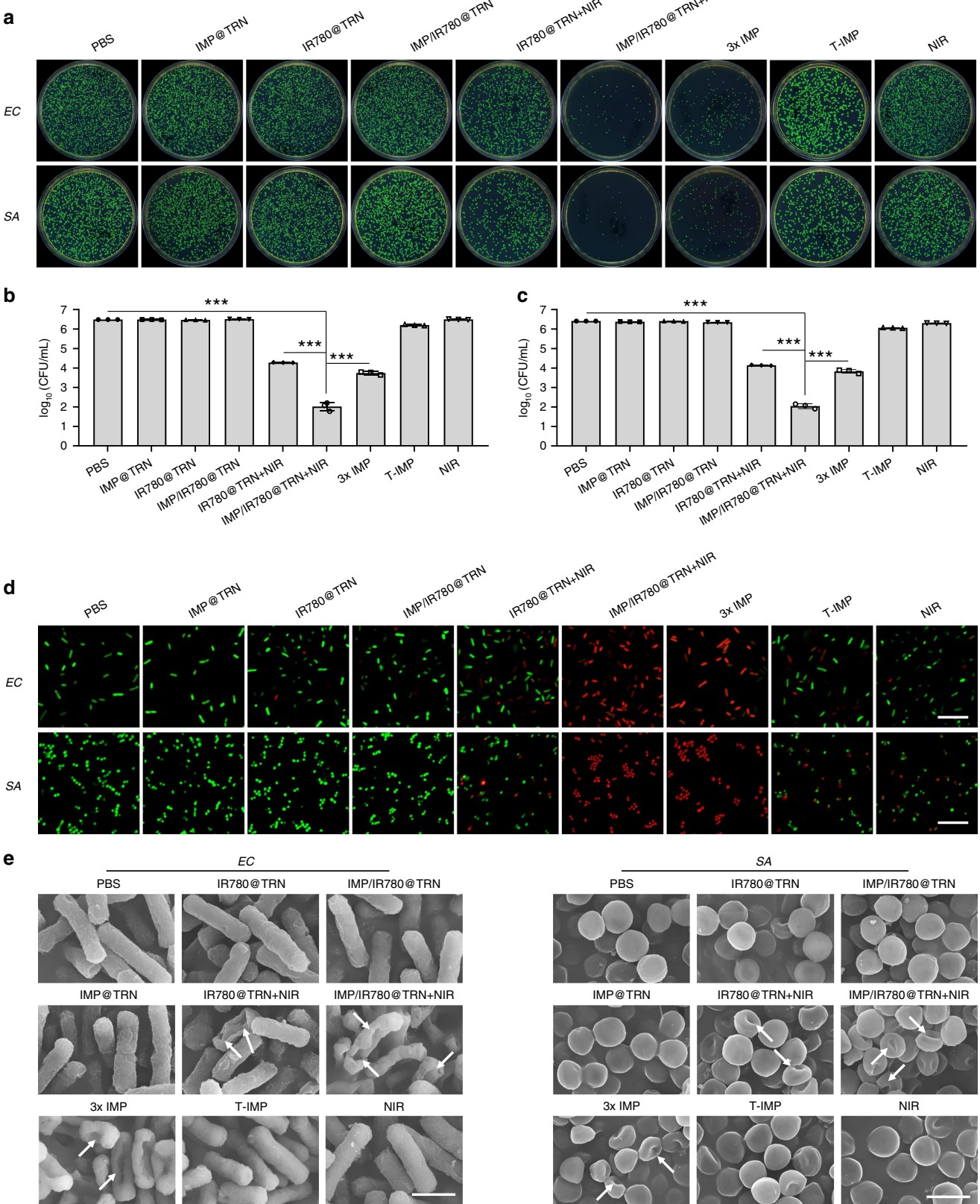

**Fig. 4** In vitro antibacterial activities of TRIDENT against antibiotic-sensitive bacteria. **a** Images of the colonies formed on Luria-Bertani broth-agar plates (*E. coli*, EC, and *S. aureus*, SA). Statistical analysis of the bacterial cell viability by $\log_{10}$ (CFU mL$^{-1}$) for **b** EC and **c** SA. Data are presented as mean ± s.d. ($n = 3$). $**p < 0.01$, $***p < 0.001$ (Student's $t$ test). **d** Images of live (green fluorescence) and dead (red fluorescence) bacterial cells following various treatments. Scale bars: 10 μm. **e** SEM images of bacteria after treatment with different groups. White arrows denote the morphological damages in bacterial cell. Scale bars: 1 μm. Source data are provided as a Source Data file

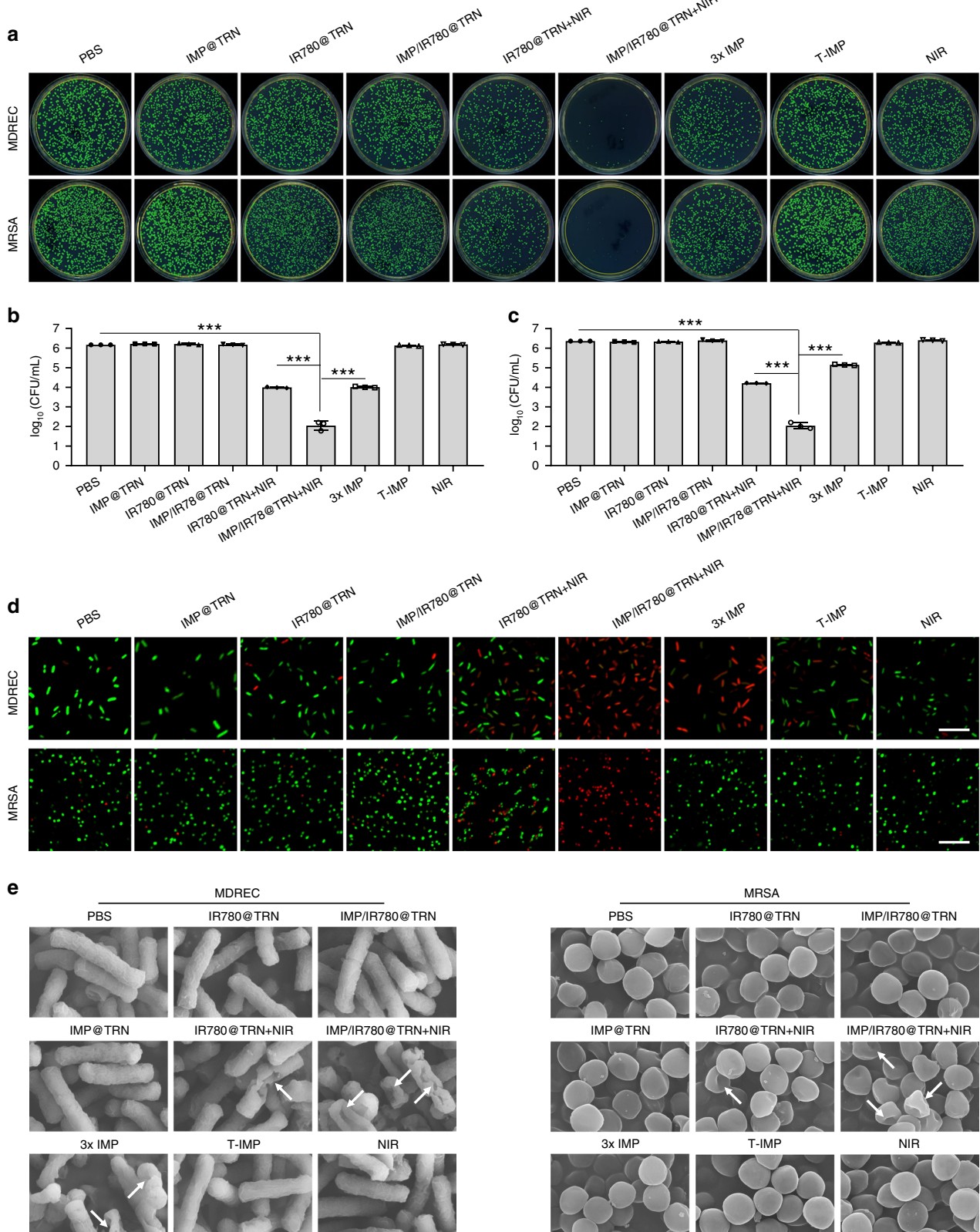

**Fig. 5** In vitro antibacterial activities of TRIDENT against clinical multidrug-resistant (MDR) bacteria (MDREC and MRSA). **a** Images of the colonies formed on Luria-Bertani broth-agar plates. Statistical analysis of the bacterial viability by $\log_{10}$ (CFU mL$^{-1}$) for **b** MDREC and **c** MRSA. Data are presented as mean ± s.d. ($n = 3$). **$p < 0.01$, ***$p < 0.001$ (Student's $t$-test). **d** Images of live (green fluorescence) and dead (red fluorescence) bacterial cells following various treatments. Scale bars: 10 μm. **e** SEM images of bacteria after treatment with different groups. White arrows denote the morphological damages in bacterial cell. Scale bars: 1 μm. Source data are provided as a Source Data file

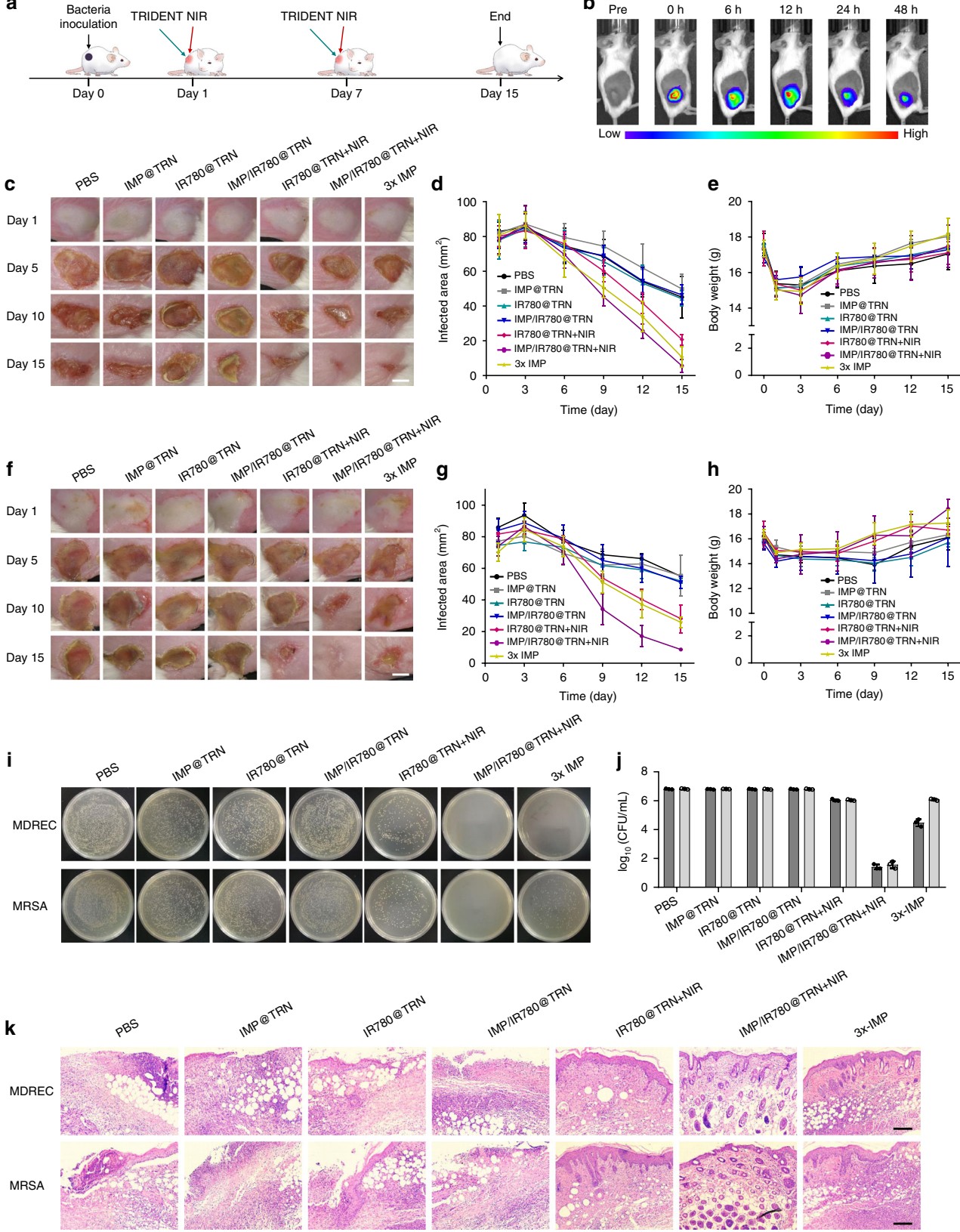

NIR-triggered nanomaterial complex (TRIDENT) for effective bacterial killing with three functions. Firstly, fluorescent IR780 molecules enable real-time monitoring of the amount of the complex at the infection site, guiding the use of NIR irradiation at

the best time. Secondly, NIR-irradiated IR780 molecules will generate a thermal response that not only melts the TRN but also damage the cell membrane via photothermal effect, which will assist with the third function for reducing the resistance to IMP

**Fig. 6** Antibacterial activities of TRIDENT in vivo. **a** Schematic diagram of the experimental process. **b** Fluorescence images showing the retention of TRIDENTs in the infected skin after the local injection. The high fluorescence intensity lasts for a long time at the infected site. **c–e** Photographs of the MDREC-infected skin of mice during treatment with the different formulations for 15 days, together with the size of the infected area and the body weights of the mice. Scale bar: 5 mm. **f–h** Photographs of the MRSA-infected skin of mice, and quantification of the infected area and animal body weights. Scale bar: 5 mm. Data are presented as mean ± s.d. ($n = 3$). **i** Photographs of plated bacterial colonies obtained from infected skin tissues of mice in the seven treatment groups. **j** Corresponding statistical analysis of the bacterial viability. Data are presented as mean ± s.d. ($n = 3$). **k** Histological photomicrographs of skin tissue sections of infected mice after completion of the in vivo antibacterial activity experiment. Scale bars: 20 μm. Source data are provided as a Source Data file

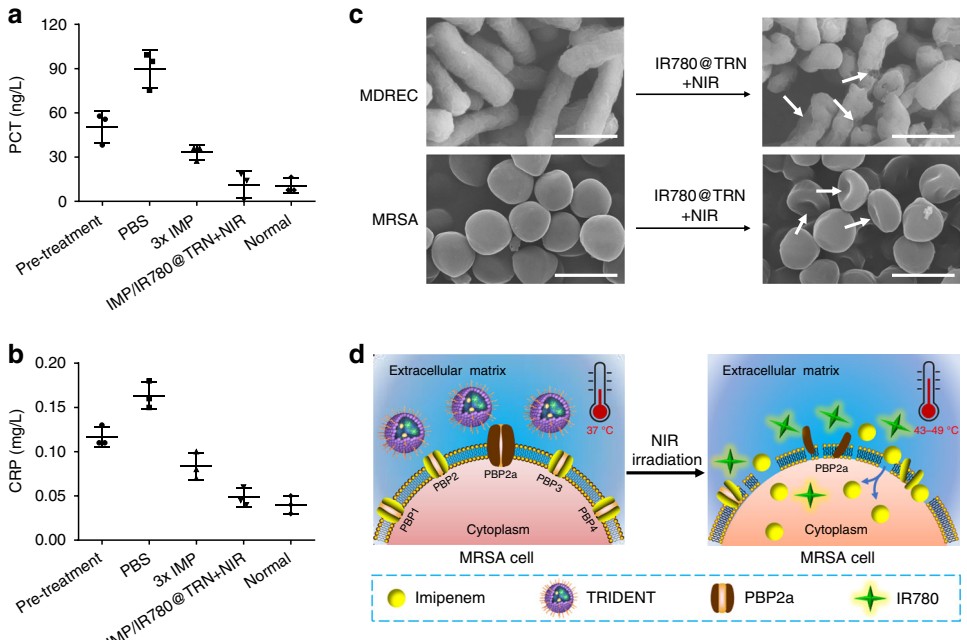

**Fig. 7** The sterilization mechanism of TRIDENT. **a** Levels of procalcitonin (PCT) and **b** c-reactive protein (CRP) in blood samples from MRSA-infected mice in different treatment groups and healthy (non-infected) mice. Data are presented as mean ± s.d. ($n = 3$). The control group was treated with PBS. **c** SEM images of bacteria before and after treatment by IR780@TRN + NIR. Scale bars: 1 μm. **d** Scheme of the killing process on MRSA by TRIDENT under NIR irradiation. PBP penicillin-binding protein. Source data are provided as a Source Data file

and promoting the rapid diffusion of IMP into cytoplasm to chemically interfere with bacterial cell wall synthesis. Both antibiotic-sensitive and clinical MDR bacteria can be killed in vitro (Figs. 4, 5) and in vivo (Fig. 6, Supplementary Figs. 12–15) via the highly effective chemo-photothermal therapy. The highly effective combinational antimicrobial activity may help to avoid the further production of drug-resistant bacteria. In addition, by encapsulation within TRN, the degradation and nephrotoxicity of IMP by renal enzyme dehydropeptidase 1, usually avoid through co-administered with cilastatin in clinic[32], will be sharply reduced when used in vivo, which will reduce the dosage of IMP and enhance the biosafety (Supplementary Fig. 16)[44,45].

It is notable that NIR-irradiated TRIDENT killed MRSA, frequently resistant to β-lactam antibiotics including IMP, with an enhanced synergistic effect compared to antibiotics or photothermal therapy alone (Figs. 4, 5, 6, and Supplementary Figs. 12–15). We deduced that several mechanisms may contribute to this powerful antibacterial activity. MRSA is resistant to β-lactam antibiotics because it has acquired the *mecA* gene, which encodes an additional penicillin-binding protein 2a (PBP2a) that blocks the action of β-lactam antibiotics[46,47]. IMP is able to kill some clinically isolated MRSA strains in part due to the relatively low amount of PBP2a in these strains. In addition, previous reports showed that the combination of IMP and vancomycin can

enhance the ability of IMP to kill MRSA, which may be attributed to vancomycin-induced destruction of the membrane and PBP2a[48,49]. These results indicate that destroying or denaturing PBP2a on the MRSA surface may increase the activity of IMP against MRSA. In our design, NIR-irradiated IR780 molecules can convert light energy into heat to mediate the lysis of MRSA by denaturing the bacterial membraneous proteins/enzymes and disturbing their membrane integrity (Fig. 7c), which will cause irreversible damage to the structure of the bacterial cells[50,51]. The damaged structures would in turn accelerate the permeation of IMP into the bacterial cytoplasm, which would then lead to impaired bacterial cell wall synthesis and finally destroyed MRSA (Fig. 7d).

The proposed TRIDENT system demonstrated potential in treating MDR bacterial infections, which is particularly useful in preventing further infection-associated deterioration. Our results clearly showed that the locally infected mice developed serious bloodstream infection or sepsis if no treatment was applied. Medication with a low-dose antibiotic (IMP) can partially alleviate this process. Fortunately, all the mice were completely healed after NIR-irradiated TRIDENT treatment for 15 days. These results indicate that NIR-irradiated TRIDENT is highly effective at preventing the progression of skin infection and avoiding infection relapse or even deterioration. Considering that

encapsulating the antibiotic and photosensitizer into a thermal-responsive nanotransporter provides good biosafety and prevents premature release of IMP, the TRIDENT system has potential for further development into a clinical agent to prevent serious bacterial infections.

Some defects or limitations are still existed in this study. Firstly, the drug-loading efficiency could be further improved although they are comparable to some existing carriers. Secondly, more accurate antibacterial mechanisms are needed to be further elucidated. Thirdly, the strategy is inefficient in treating deeper tissue infections or systematic blood infections owing to the limited tissue penetration capacity and irradiation area of NIR light[8,13,52]. Therefore, developing strategies employing other photosensitive molecules excited by NIR-II region light and encapsulating additional functional molecules could further improve the potential clinic application of the as proposed TRIDENT strategy.

In summary, a synergistic antibacterial strategy that integrates the advantages of fluorescence monitoring, physical damage via photothermal therapy, and chemical inhibition derived from antibiotics has been proposed. The resultant TRIDENT system is homogeneous in size, has good biocompatibility and possesses tunable thermo-responsive characteristics. It effectively circumvents premature release of IMP and has a long retention time at infected sites. Both in vitro and in vivo tests validated that our strategy can effectively kill antibiotic-sensitive bacteria and clinical MDR bacteria at a low antibiotic dose, thus helping to reduce the adverse effects associated with high antibiotic doses. In vivo, TRIDENT decreased the possibility of sepsis and promoted the rehabilitation of infected site. Through the encapsulation of other antibiotics, the proposed antibacterial strategy can be further developed into a universal antimicrobial platform.

## Methods

**Materials**. Stearic acid (95%) and lauric acid (97%) were separately purchased from Merck (Germany) and Heowns (Tianjin, China). Imipenem (IMP, 98%) were purchased from Yuanye (Shanghai, China). IR780 iodide (IR780, 98%) and soybean lecithin (Mw ~750) were purchased from Sigma (USA). 1,2-distearoyl-sn-glycero-3-phosphoethanolamine-N-[methoxy (polyethylene glycol)-2000] (Mw = 2000, DSPE-PEG2000) was purchased from Lipoid (Germany).

**Preparation of the IMP/IR780 thermo-responsive nanostructure (TRIDENT)**. The TRIDENT nanoparticles were fabricated through a reported method with some modification[24]. Briefly, the TRN solution was made from lauric acid and stearic acid at a mass ratio of 3.5:1, and the final concentration was 4 mg mL$^{-1}$, then droplets of TRN stock solution containing IMP and IR780 were added into 4% aqueous solution of phospholipids formed by lecithin and DSPE-PEG2000 at 55 °C followed by vigorous vortexing for 5 min. The solidification of the nanoparticles was realized through a rapid chilling process with ice water. IMP/IR780@TRN was finally obtained after washing with water and purifying by ultracentrifugation at room temperature. A similar procedure was utilized to prepare IMP@TRN and IR780@TRN.

**Characterization of TRIDENT**. The phase change temperatures of TRN, IMP@TRN, IR780@TRN and IMP/IR780@TRN were measured by differential scanning calorimeter (Perkin Elmer, Diamond). The size and morphology of IMP@TRN, IR780@TRN and IMP/IR780@TRN were characterized by dynamic light scattering (Malvern, Zetasizer Nano ZS) and transmission electron microscopy (HITACHI, Ht-7700), respectively. The amount of IMP and/or IR780 in IMP@TRN, IR780@TRN and IMP/IR780@TRN was determined by UV-vis spectrophotometer (Mapada Model UV-1800 spectrophotometer) according to a calibration curve at 294 nm and 780 nm in methanol aqueous solution (V/V = 1:99), respectively.

**Photothermal properties of TRIDENT**. IMP/IR780@TRN was dispersed in PBS at five concentrations (0, 5, 10, 20, 40 μg mL$^{-1}$ for IR780). Then 1 mL solution was added to a cuvette and irradiated with an 808 nm NIR laser source at a power density of 0.5 W cm$^{-2}$ for 5 min. The temperatures and photographs of the NIR-irradiated IMP/IR780@TRN solution were recorded at 20 s intervals using an infrared imaging device (Fluke, Ti400). To obtain the accurate morphology of TRIDENT when the irradiation just stopped, the TRIDENT solution was immediately immersed in liquid nitrogen to be rapidly solidified. After cooling down in liquid nitrogen for 1 min, the solidified solution was naturally warmed up to ambient temperature. The morphology and particle size distribution of the irradiated TRIDENT were characterized by TEM and DLS, respectively.

**Light-triggered IMP release of TRIDENT**. 1 mL TRIDENT solution (500 μg mL$^{-1}$ for IMP) were separately added to four dialysis tubes (MWCO = 3500D) and then were shaken in an orbital shaker. NIR irradiation was applied to the four tubes for 5, 10, 15, and 20 min in turn at the time point of 3 h, and then vibration again. At selected time intervals (0, 1, 2, 4, 6, 8, 10, 12, and 24 h), 500 μL released medium were retrieved and the absorbance at 294 nm was measured by high performance liquid chromatography (SHIMADZU, LC-20A) to determine the amount of the released IMP according to the calibration curve. The cumulative release was defined as: mass of released IMP/mass of loaded IMP × 100%.

**Cytotoxicity of TRIDENT**. The cytotoxicity of IMP/IR780@TRN was investigated using a cell counting kit-8 (CCK-8) assay. 3T3 cells and HUVECs (purchase from the Institute of Basic Medical Sciences) were seeded in 96-well plates (5000 cells per well). After incubation overnight, 100 μL serial concentrations of IMP/IR780@TRN were added into the wells, which were then incubated for 24 h. Finally, 10 μL CCK-8 solution was added into each well with another 2 h incubation. The optical density (OD) at a wavelength of 450 nm (OD$_{450}$) was measured with a microplate reader (Perkin Elmer, EnSpire). Cells incubated in DMEM were used as control.

**Bacterial culture**. E. coli (ATCC 25922), S. aureus (ATCC 25923) and clinically isolated MDREC or MRSA were cultured in Luria–Bertani (LB) broth medium in a shaking incubator (200 r.p.m.) at 37 °C and harvested at the logarithmic growth phase by centrifugation at 4500 g for 5 min. After washing with PBS three times, the bacteria were resuspended in PBS for further use. The concentration of bacteria was monitored by measuring the optical density at 600 nm (OD$_{600}$) through microplate reader. To investigate the antibacterial effect of our TRN, seven experimental groups, PBS, IMP@TRN, IR780@TRN, IMP/IR780@TRN, IR780@TRN + NIR, IMP/IR780@TRN + NIR, 3× IMP, T-IMP, and NIR, were established. And the group treated with PBS was the control group.

**In vitro antibacterial experiments**. The bacteria were diluted to the appropriate level ($1 \times 10^5$ to $1 \times 10^6$ CFU mL$^{-1}$) and then treated with different formulations for 2 h at 37 °C on a shaking incubator (200 r.p.m.). The groups of IR780@TRN + NIR, IMP/IR780@TRN + NIR, and NIR were separately irradiated with an 808 nm laser at a power density of 0.5 W cm$^{-2}$ for 5 min. Then, the solution (50 μL) of the bacterial suspension was spread on LB plates followed by 100× dilution of the solution, and the number of CFU in each plate was counted (Shineso, Supcre G9) and calculated by the form of $\log_{10}$ CFU mL$^{-1}$ to decide the antibacterial effect of different conditions after cultivation for 24 h at 37 °C ($n = 3$). 200 μL bacterial suspension ($1 \times 10^6$ to $1 \times 10^7$ CFU mL$^{-1}$) treated with different formulations were seeded in 96-well plates ($n = 3$) and cultured for over 48 h on a shaking incubator (200 r.p.m.) at 37 °C. The OD$_{600}$ of each well was monitored at different time points during the process.

**In vitro thermal response of bacteria**. 1 mL bacterial suspension ($1 \times 10^5$ to $1 \times 10^6$ CFU mL$^{-1}$) were separately added into 5 tubes and shaken at 37 °C for 2 h. Then four of five tubes were separately shaken at 40 °C, 43 °C, 46 °C, and 49 °C for 5 min. Next, the solution (50 μL) of the bacterial suspension in tubes was spread on LB plates with 100× dilution of the solution, and the number of colony-forming units (CFU) in each plate was counted and calculated by the form of $\log_{10}$ (CFU mL$^{-1}$) to decide the antibacterial effect of different conditions after cultivation for 24 h at 37 °C ($n = 3$). Another 1 mL bacterial suspension ($1 \times 10^5$ to $1 \times 10^6$ CFU mL$^{-1}$) were separately added into four tubes and then separately shaken at 37 °C for 2 h. Three of four tubes were separately shaken 1 min, 3 min, and 5 min at 49 °C. Then follow the same steps as before.

**Morphology of bacteria after treatment**. Different treatments were added into the bacterial suspension ($1 \times 10^5$ to $1 \times 10^6$ CFU mL$^{-1}$) at a final concentration of 20 μg mL$^{-1}$ (in terms of IMP), and the suspension treated with IR780@TRN + NIR, IMP/IR780@TRN + NIR and NIR alone were irradiated with an 808 nm NIR laser at a power density of 0.5 W cm$^{-2}$ for 5 min after incubation for 2 h at 37 °C. For SEM, the bacterial suspensions were centrifuged at $4500 \times g$ for 10 min and then washed three times with PBS. Then the collected bacterial cells were fixed with 2.5% glutaraldehyde overnight at 4 °C. After washing with PBS three times, the bacterial cells were dehydrated through sequential treatments of 30%, 50%, 70%, 80%, 90%, 95%, and 100% ethanol for 30 min. The final samples were dried in vacuum freezing drying oven before being inspected by SEM (HITACHI, S-4800).

**Live/dead staining assay**. The viability of bacteria in the test samples was qualitatively assessed using a Live/Dead® Baclight™ bacterial viability kit (Life Technologies, Carlsbad), and all experiments were carried out according to the manufacturer's instructions. Bacteria ($1 \times 10^6$ to $1 \times 10^7$ CFU mL$^{-1}$) were separately treated with the seven formulations (20 μg mL$^{-1}$, in terms of imipenem) for 2 h and collected by centrifugation at $10000 \times g$ for 3 min. After washing with 0.9% NaCl, the bacteria were stained with 1.5 μL of 3.34 mM green-fluorescent nucleic

acid stain (SYTO 9) and 1.5 μL of 30 mM red-fluorescent nucleic acid stain (PI) for 15 min. Bacterial samples were then imaged using a laser scanning confocal microscope (ZEISS, LSM710).

**Mouse infection models.** All animal experiment protocols were reviewed and approved by the Institutional Animal Care and Use Committee of National Center for Nanoscience and Technology and complied with all relevant ethical regulations. 6- to 8-week-old female BALB/c mice purchased from Beijing Charles River Company and raised in a specific pathogen-free grade laboratory were used to evaluate the in vivo antibacterial activity and toxicity of the TRN. The infection model was established by subcutaneously inoculating clinically isolated MDREC or MRSA ($1 \times 10^6$ to $1 \times 10^7$ CFU mL$^{-1}$, 100 μL) into the right flank of the mice. For real-time fluorescence imaging of living animals, infected mice were subcutaneous injected with IMP/IR780@TRN at an equivalent IR780 dose of 1.0 mg kg$^{-1}$. The fluorescence images were recorded using an IVIS Spectrum Imaging System (Perkin Elmer) at selected time points. For the in vivo photothermal test, 100 μL IMP/IR780@TRN (1 mg kg$^{-1}$ of IMP) were subcutaneously injected into the infected skin of MRSA-infected mice first and then irradiated by 808 nm laser at the power density of 0.5 W cm$^{-2}$ for 5 min ($n = 4$). The temperatures of infected skin were recorded by an IR camera (Fluke, Ti400).

**In vivo antibacterial experiments.** Mice were randomly divided into seven groups 24 h after bacterial inoculation and 100 μL PBS, IMP@TRN (1 mg kg$^{-1}$ of IMP), IR780@TRN (1 mg kg$^{-1}$ of IR780), IMP/IR780@TRN (1 mg kg$^{-1}$ of IMP), IR780@TRN + NIR (1 mg kg$^{-1}$ of IR780), IMP/IR780@TRN + NIR (1 mg kg$^{-1}$ of IMP), 3× IMP (3 mg kg$^{-1}$), and IMP (0.4 mg kg$^{-1}$) were subcutaneously injected twice, on days 1 and 7, NIR laser was applied to group of IR780@TRN + NIR, IMP/IR780@TRN + NIR after the two subcutaneous injections. Same irradiation was also applied to the group of NIR. Any treatments were performed 24 h after the inoculation ($n = 3$). Afterwards, we dynamically monitored the healing processes of the infected areas through macroscopic photographs, and damaged areas and body weights of the infected mice were measured throughout the whole treatment period (15 days). In order to assess the in vivo therapeutic effect of the different formulations, the infected skin tissues were collected and fixed in 4% fixative solution, then stained with hematoxylin and eosin (H&E), and finally examined using a digital microscope. To determine the number of bacteria at the infection site, the infected tissues were homogenized in PBS (1 mL) and the suitably diluted tissue solutions were plated on LB agar plates. And the bacterial colony were counted after the incubation for 24 h at 37 °C (three duplicates).

**Analysis of sepsis biomarkers.** To estimate the levels of biomarkers associated with sepsis, mice were randomly divided into four groups 24 h after bacterial inoculation. Before treatment, one group of the infected mice and normal mice were sacrificed and the blood was collected for further analysis. Next, the three groups were separately treated with PBS, free IMP and IMP/IR780@TRN + NIR by two subcutaneous injections on days 1 and 7. Healthy mice (non-infected) were used as negative controls. At the end of the treatment period (15 days), blood was collected from the infected mice and the levels of pro-calcitonin and c-reactive protein were measured by enzyme-linked immuno-sorbent assay kits (Andy Gene, Beijing) and auto biochemistry analysis (Roche, Cobas e411). The results were compared with blood from healthy (non-infected) mice ($n = 3$).

**In vivo biosafety assessment.** To verify the biosafety of IMP/IR780@TRN, mice were randomly divided into three groups and treated with PBS, IMP/IR780@TRN + NIR and free IMP by two subcutaneous injections on days 1 and 7. On day 15, the mice were sacrificed and the blood was collected by retro-orbital bleeds and centrifuged at $800 \times g$ for 20 min. The blood serum samples were analyzed for two renal function markers, blood urea nitrogen (BUN), and creatinine (CREA). Histological analysis of five major organs (heart, liver, spleen, lung, and kidney) was carried out by H&E staining.

**Statistical analysis.** Graphs were plotted and appropriate statistical analyses were conducted using GraphPad Prism 5.0 (*$p < 0.05$, **$p < 0.01$ and ***$p < 0.001$). Data were calculated and processed as mean ± s.d. Comparison between groups were analyzed with one-way analysis of variance (ANOVA).

**Reporting summary.** Further information on research design is available in the Nature Research Reporting Summary linked to this article.

## Data availability

The main data supporting the findings of this study are available within the article and its Supplementary Information. Extra data are available from the corresponding author upon reasonable request. The source data underlying Figs. 2c–e, 3a, 3c, 3f, 4b, 4c, 5b, 5c, 6d, 6e, 6g, 6h, 6j, 7a, 7b as well as Supplementary Figures 1, 3–8, 11–12, 14, 16 are provided as a Source Data file.

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

## Acknowledgements

This work was financially supported by the National Natural Science Foundation of China (Grant Nos. 81871733, 81572079, 81601845, 31630027, and 81601603) and NSFC-DFG project (31761133013).The authors also appreciate the support by the external cooperation program of the Chinese Academy of Science (121D11KYSB20160066), the "Strategic Priority Research Program" of the Chinese Academy of Sciences (XDA09030301), Natural Science Foundation of Chongqing (CSTC2018JSCX-MSYB1033), the Fundamental Research Funds for the Central Universities (2018CDGFSG0017, 2019CDYGZD005, and 2019CDYGZD007).

## Author contributions

G.Q., W.G., Y.L. and X. Liang conceived and designed the project. G.Q., X.Z., N.G., J.C., X. Li, Y.W., Z.Z. and Y.Z. performed the experiments. G.Q., X.Z., N.G., J.C., X. Li, Y.G., Y.W., W.G., Y.L. and X. Liang collected, analysed and interpreted the data. W.G., Y.L. and X. Liang supervised the project. G.Q., W.G., Y.L. and X. Liang wrote the manuscript. All authors discussed the results and commented on the manuscript.

## Additional information

**Competing interests:** The authors declare no competing interests.

