## [Peer Review File · Nature Communications]

Reviewers' comments:

Reviewer #1 (Remarks to the Author):

In this study, the authors developed an antibiotic (IMP) and photothermal agent (IR780) loaded thermo-responsive phase-changeable drug delivery nanoparticle (TRIDENT) system for chemo-photothermal therapy of bacterial infection. The as-developed TRIDENTs were well characterized by TEM, DLS, DSC and UV-Vis spectroscopy, and their thermo-responsive properties were elucidated using laser irradiation. The antibacterial properties of TRIDENTs upon laser irradiation were demonstrated using in vitro and in vivo experiments. The obtained experimental results revealed that the developed TRIDENTs could effectively eradicate the drug-sensitive and drug-resistant bacteria chemo-photothermally upon laser irradiation. The concept of this study is not novel. The use of the thermo-responsive nanocarriers for chemo-photothermal therapy of bacterial infection has been extensively studied previously. Also, the utilization of eutectic mixture of lauric acid and stearic acid (phase-change material) for temperature responsive drug delivery has also been reported elsewhere. Further, the TRIDENTs that developed in this study required two time injection and irradiation for the efficient in vivo antibacterial eradication, limiting their practicality. Additional comments/suggestions are shown below.

1. What was the PDI of the as-developed TRIDENTs in DLS measurements?
2. Figure 1b. The quality of the TEM image is poor. Please provide a better TEM image containing nanoparticles with uniform size.
3. Figure 1d. The characteristic peak of IMP in TRIDENTs is not obvious. Additionally, it is suggested that the authors include a control group of blank nanoparticles in the study.
4. Figures 2 d and e. The authors claimed that the morphology of TRIDENTs did not change before and after irradiation. How could this happen as the TRIDENTs are thermo-responsive? What were the initial and final temperatures of the test solution before and after laser irradiation?
5. Supplementary Figure 1. The authors presented the morphologies of the IMP TRN before and after irradiation. As the IMP TRN did not contain any PTT agent, how could the authors irradiate the IMP TRN?
6. At what temperature the authors performed laser irradiation for the in vitro antibacterial experiments? RT or 37 °C? What is the final temperature of the test solution that could be obtained after laser irradiation?
7. What was the mechanism of using free IMP in killing MDREC but not MRSA?
8. Figures 3d and 4d. The quality of the confocal images is poor.
9. Figures 3e and 4e. The figure captions indicated "TEM". Should it be "SEM"?
10. According to Figure 5b, the TRIDENTs were present at the infection site even after 48 h post injection. What could be the reason for this observation? What would happen if the tested particles were injected into the healthy skin? Generally, if there is no interaction between the injected nanoparticles and the local tissue, they should drift away from the injection site immediately.
11. It is suggested that the authors perform in vivo photothermal experiments for their developed TRIDENTs.
12. Figure 6c. A typo of "imagines" as found.
13. Page 22, Line 11. The meaning of the sentence "the solution in one tube.....for 24 h" is confusing.

Reviewer #2 (Remarks to the Author):

This original work by Qing et al. proposes the use of drug and fluorophore loaded liposomes for use as a combinatorial therapy against bacterial infections. They have dubbed their system TRIDENT, and I applaud their efforts to make an acronym work, even if it's a little shaky. Laser application is used to heat the fluorophore, which disrupts the liposomal membrane, releasing an antibiotic in a local fashion. This method purports that the photothermal heating enhances bacterial killing and the local

antibiotic release helps to reduce toxicity. On the whole, the work is very interesting, and shows promise. The use of the English language is acceptable, although some of the abbreviations probably need to be defined sooner (or at all). However, the authors omit several essential controls from their study that would elucidate the mechanisms at play. Additionally, there are several minor issues that need to be addressed. My concerns are listed below in no particular order. I recommend the authors revise the work and resubmit once essential changes have been made.

The essential controls that must be added to make this manuscript complete are:

- A) Viability of bacteria in response to thermal stress akin to what is demonstrated with the TRIDENT system (e.g. +12 C for 5 min).
- B) There needs to be an IMP treatment group based on the theoretical released IMP dose rather than 3x the MIC, which is a relatively arbitrary value.
- C) All thermal treatments are TRIDENT, there needs to be a control thermal group that uses the TRIDENT platform without IMP.
- D)

1) Since the G1-G7 groups are referenced heavily in the figures and the text, a table describing these groups would be extremely helpful.

2) The hydrodynamic and core diameters need to be more precisely defined that " \sim 40 nm" particularly if the size distribution is "within 50 - 80 nm." Specific values and standard deviations (or SEMs) are essential. This is particularly important since later remarks are made regarding the percentage of particles in the 80-100 nm size range.

3) Similarly standard deviations are needed for the loading and encapsulation efficiency. It would also be helpful for the authors to define these terms precisely in the text where relevant. I would also caution the authors to pay attention to significant figures when standard deviations are added. In general, the authors need to include precise numbers throughout the results section.

4) Figure 5 indicates that the fluorescence is the result of tail vein injections (implying some form of selective localization at the site of infection), which is NOT described in the methods (subcutaneous) or main text. If this is an error it needs to be corrected.

5) At least one citation should be included with the claim that 12 C "could lead to serious damage to cell walls". Expanding on this, it would be extremely beneficial for the authors to show the response of these bacteria to various temperatures and times that are relevant to the study.

6) The inclusion of the DSC data is a welcome change from most liposome studies. However, it would be extremely valuable to include data from drug and/or dye loaded TRIDENT to confirm that the thermal properties are not changed as a result of these new contributions.

7) The introduction should provide a more thorough literature review regarding combinatorial PTT and antibiotic therapy. This is an emerging area of research, and the various approaches should be addressed. In particular, use of other nanomaterials, such as graphene, gold, and silver should be addressed given their inherent photothermal capacity.

8) The bacterial viability data should be presented in a log scale rather than a percent. It is not instructive in a situation where $1e6$ cells are initially treated to show that $<1\%$ CFUs remain. There can be literally thousands of cells remaining and an additional day of culturing may show that valuable therapeutic effect has been realized.

9) Is IR780 being used as a fluorescent tracker/photothermal transducer or is there an additional purpose as a photosensitizer?

10) Does colloidal stability simply mean the hydrodynamic diameter did not change? More importantly, are there other changes, like leakage, that occur during the process?

11) The release profile is unimpressive. In 24 hours roughly 15% of the loaded drug leaks out, while laser treatment increases the release from 15% to a whopping 30%. Why is this thermally triggered release so meager? Where is the rest of the drug? By extension, the schematic is misleading as it suggests that the liposome will be fundamentally destroyed at the elevated temperature (indicating release on the order of 100%). So it is probably wise to revise the schematic as well. The authors do comments that the TRNs are partially damaged from the photothermal treatment, but the extent of this damage and the underlying implications are left unstated. Additionally, the loading of IMP seems extremely low.

12) Fig4e shows similar membrane damage in quite a few panels more than is discussed in the text.

13) I appreciate the addressing of biosafety, but the authors should make clear that this is an acute rather than chronic study.

14) It is inadequate to use a methanol calibration in a aqueous system without justifying their equivalence.

15) The discussion is more or less useless. It's a summary of the findings that provides no real insight into the platform nor does it discuss the limitations or potential improvements.

Reviewer #3 (Remarks to the Author):

This manuscript describes the phase changeable fatty acid-based nanotransporter for bacteria therapy. The system consisted of fatty acid mixture of lauric acid (LA) and stearic acid (SA) at 3.5:1 in weight, that can be melted in range of 43 ~ 44 oC. The mixture was used as a gate material for on-demand release of pre-loaded antibacterial agent without undesired passive diffusion of the agent. Addition of IR780, a photothermal agent capable of converting light energy into heat, mediated bacterial lysis by denaturing the bacterial membrane proteins. Consequently the chemo-photothermal effect could successfully kill the bacteria. Although the synergistic therapy using phase changeable fatty-acid based nanoparticles has been already reported by Younan Xia and coworkers, especially for cancer therapy, this work is very impressive in that the use of fatty acid-based nanoparticles was achieved for the treatment of bacterial infection. Besides, the biosafety of the nanoparticles was investigated using in-vivo mouse model, which has not been reported, to the best of my knowledge, in previous studies. The experimental design and data of this study are clear to convince to the conclusion. I strongly support the publication of this work after several things are clarified.

1. According to previous literatures studying on the mixture of LA and SA (J. Chem. Eng. Data 2011, 56, 2889; Macromol. Mater. Eng. 2016, 301, 887), the mixture has only one melting point (called eutectic point) at a specific composition. In the other forms, the mixture has another melting DSC peak as well as the eutectic peak. However, in this work, each composition of LA and SA had one melting point, as shown in Figure S2. The authors need to explain this result more detail.

2. The authors irradiated NIR light to TRIDENT for studying on the morphology of the system and its anti-bacteria ability. When the light was exposed, what was the temperature of the system ?

3. The authors stated the photothermal effect of IR780 in response to NIR light melted TRN. But, as shown in Figure 2d and 2e, there was no significant change in the morphology and size of the system and the diminutive particles were formed. If the melting completely occurred, I think the formation of the much more diminutive particles should be observed due to the destruction of the fatty-acid based

nanoparticles, as illustrated in Figure 2g. Why was the morphology and size of the TRN unchanged after NIR irradiation? Was it attributable to non-uniform distribution of IR780 within TRN, leading to partial damage ?

4. When the release of IMP from TRIDENT occurs under NIR light, it seems that IR780 can be released as well. How much amount of the photothermal agent was released?

5. In bacterial test, the treatment with only NIR irradiation need to be demonstrated in order to clarify the anti-bacterial ability of TRIDENT + NIR.

Minor Issues

1. Figure 3e, 4e shows SEM images not TEM images.

Response to Reviewers point-by-point:

Reviewer #1:

In this study, the authors developed an antibiotic (IMP) and photothermal agent (IR780) loaded thermo-responsive phase-changeable drug delivery nanoparticle (TRIDENT) system for chemo-photothermal therapy of bacterial infection. The as-developed TRIDENTs were well characterized by TEM, DLS, DSC and UV-Vis spectroscopy, and their thermo-responsive properties were elucidated using laser irradiation. The antibacterial properties of TRIDENTs upon laser irradiation were demonstrated using *in vitro* and *in vivo* experiments. The obtained experimental results revealed that the developed TRIDENTs could effectively eradicate the drug-sensitive and drug-resistant bacteria chemo-photothermally upon laser irradiation. The concept of this study is not novel. The use of the thermo-responsive nanocarriers for chemo-photothermal therapy of bacterial infection has been extensively studied previously. Also, the utilization of eutectic mixture of lauric acid and stearic acid (phase-change material) for temperature responsive drug delivery has also been reported elsewhere. Further, the TRIDENTs that developed in this study required two time injection and irradiation for the efficient *in vivo* antibacterial eradication, limiting their practicality. Additional comments/suggestions are shown below.

Response: We highly appreciate the reviewer putting forward pertinent suggestions for our study. Actually we fabricated the TRIDENTs according to the methods proposed by Prof. Younan Xia group with slight modification¹. Different from other studies on cancer treatment, this study provides a chemo/thermally synergistic method for bacterial infection. We are excited to find that the photothermal therapy promotes the killing of MRSA, a known bacteria resistant to β -lactam antibiotics such as IMP, which hasn't been reported by others. Besides, all the revised contents have been marked red and the point-by-point responses are listed below.

1. What was the PDI of the as-developed TRIDENTs in DLS measurements?

Response: Thanks for your comment. The hydrodynamic diameters of the as-prepared TRIDENT were recorded by DLS based on three independent measurements. The as-prepared TRIDENT showed average hydrodynamic diameters of 63.39 ± 7.75 nm with PDI of 0.27 ± 0.05 . The data have been added in the revised manuscript (Page 6, line 96-99).

2. Fig. 1b. The quality of the TEM image is poor. Please provide a better TEM image containing nanoparticles with uniform size.

Response: Thanks for your helpful comment. According to your suggestion, we optimized the preparation of TRIDENTs very carefully. The obtained TRIDENTs (IMP/IR780@TRN), IR780@TRN, and IMP@TRN were subjected to TEM observation, which showed uniform size and spheroid morphology. The TEM images with improved quality have been added in Fig. 1b (Page 7, line 109) and Supplementary Fig. 1.

Supplementary Fig. 1. TEM images and particle size distribution histograms of IMP/IR780@TRN, IR780@TRN, and IMP@TRN. a, b) IMP/IR780@TRN. c, d) IR780@TRN. e, f) IMP@TRN. Scale bars: 200 nm.

3. Fig. 1d. The characteristic peak of IMP in TRIDENTs is not obvious. Additionally, it is suggested that the authors include a control group of blank nanoparticles in the study.

Response: Thanks for your careful suggestion. For strict comparison, we re-performed the absorption spectrum of TRIDENT, Vehicle, IMP, and IR780 at equal concentrations of corresponding components (Vehicle, IMP, or IR780). The characteristic peak at 294 nm was observed on both the spectra of IMP and TRIDENT, while no obvious peak was found on the spectra of vehicle in Fig. 1d. (Page 7, line 110). The revised Fig. 1d is also listed as below.

Fig. 1d. UV-vis absorption spectra of IMP, IR780, TRIDENT and vehicle (blank nanoparticles).

4. Fig. 2 d and e. The authors claimed that the morphology of TRIDENTs did not change before and after irradiation. How could this happen as the TRIDENTs are thermo-responsive? What were the initial and final temperatures of the test solution before and after laser irradiation?

Response: In our previous test, the samples for TEM observation and DLS measurements were derived from the TRIDENT solution, which was naturally cooled down to room temperature after laser irradiation. However, the TRN could naturally solidify into nanoparticles during the slowly

cooling down process because of the intrinsic hydrophobicity of the inner core (lauric acid and stearic acid). Therefore, it is challenging for TEM observation to veritably reflect the morphology of TRIDENT immediately at the end of irradiation. Of note, to the best of our knowledge, the published literatures have barely provided and employed convinced TEM images to validate the morphology change raised from laser irradiation to date¹⁻³. Maybe it is the above proposed dilemma that hinders researchers to use TEM observation for the study.

To realize more accurate investigation on the morphology change of TRIDENT using TEM, the TRIDENT solution was immediately immersed in liquid nitrogen for rapid solidification at the end of laser irradiation. Due to the sudden decrease of the temperature, fractured TRIDENTs would immediately solidify and the instantaneous morphology of them could be frozen without any other changes. After cooling down in liquid nitrogen for 1 min, the solidified solution was naturally warmed up to room temperature (~25 °C, lower than the phase-transition temperature of the inner core). Thereby, the irradiated TRIDENTs retained their original morphology at the end of laser irradiation and suited for TEM observation (Page 23, line 380-384).

The temperature of the TRIDENTs solution was increased from ~37 °C to ~49 °C upon the laser irradiation (Response Fig. 1). Meanwhile, compared with the TRIDENTs before irradiation, many smaller nanoparticles were observed by TEM from the sample of TRIDENTs upon laser irradiation. Taking together with the decrease of hydrodynamic diameters, the morphology change suggested that the nanostructure of TRIDENTs was destroyed during the process of laser irradiation and thus thermal-responsive (Fig. 2d,e). Similar phenomenon was also found in IR780@TRN (Supplementary Fig. 4). We have revised the related contents in the updated manuscript (Page 10, line 138-142).

Supplementary Fig. 4. TEM images and particle size distribution histograms of IMP/IR780@TRN, IR780@TRN and IMP@TRN before and after the NIR irradiation. Scale bars: 200 nm.

Response Fig. 1. Thermal images of TRIDENT solution under NIR irradiation (0.5 W cm^{-2}) before and after irradiation.

5. Supplementary Fig. 1. The authors presented the morphologies of the IMP TRN before and after irradiation. As the IMP TRN did not contain any PTT agent, how could the authors irradiate the IMP TRN?

Response: Thanks for your careful comment. It is correct that no significant photothermal effect was observed because of the absence of PTT agent (IR780). The IMP@TRN was presented as the control experiment to confirm the different response between IR780@TRN and IMP@TRN under NIR laser irradiation. It showed that NIR-irradiated IMP@TRN yielded no obvious temperature changes and no phase transition of TRN was observed (Supplementary Fig. 4). Therefore, it revealed that IMP@TRN alone cannot achieve IMP release for bacterial therapy under the NIR irradiation.

Supplementary Fig. 4. TEM images and particle size distribution histograms of IMP/IR780@TRN, IR780@TRN and IMP@TRN before and after the NIR irradiation. Scale bars: 200 nm.

6. At what temperature the authors performed laser irradiation for the *in vitro* antibacterial experiments? RT or 37 °C? What is the final temperature of the test solution that could be obtained after laser irradiation?

Response: We performed laser irradiation for the *in vitro* antibacterial experiments at around 37 °C (Response Fig. 2). It would be more compatible for the next *in vivo* applications because all the injected substrates would become around 37 °C *in vivo*. The final temperature of the *in vitro* experiment upon irradiation is dependent on the laser power density, irradiation time, and the concentration of the IR780. We found that the maximum temperature of the tested solution (20 µg/mL for IR780) could reach to 49 °C after the NIR irradiation (0.5 W/cm²) for 5 min.

Response Fig. 2. Heating curves of the test solution and the thermal images of the solution at start and end point were inset in the figure.

7. What was the mechanism of using free IMP in killing MDREC but not MRSA?

Response: IMP is a kind of β -lactam antibiotics. Methicillin-resistant staphylococcus aureus (MRSA) has a special class of penicillin binding proteins (PBP2a), which have low affinity to β -lactam antibiotics. Therefore, MRSA can effectively maintain their activity for promoting the biosynthesis of bacterial cell walls even exposed to high concentrations of IMP. On the contrary, the similar penicillin-binding proteins are absent in the bacterial surface of the MDREC isolated from clinic in this study, which makes them much more sensitive to β -lactam antibiotics^{4,5}.

8. Fig. 3d and 4d. The quality of the confocal images is poor.

Response: Thanks for your comments. The confocal images in Fig. 3d and 4d have been replaced by new ones with higher quality (Page 12, line 190-191; Page 14, line 213-214). The revised Fig. was showed as below (**Response Fig. 3**).

Response Fig. 3. Images of live (green fluorescence) and dead (red fluorescence) bacterial cells following various treatments. Scale bars: 10 μm.

9. Fig. 3e and 4e. The Fig. captions indicated “TEM”. Should it be “SEM”?

Response: Yes. We have corrected this typo (Page 12, line 192; Page 14, line 215).

10. According to Fig. 5b, the TRIDENTs were present at the infection site even after 48 h post injection. What could be the reason for this observation? What would happen if the tested particles were injected into the healthy skin? Generally, if there is no interaction between the injected nanoparticles and the local tissue, they should drift away from the injection site immediately.

Response: Thanks for your insightful comments. We observed the fluorescence intensity of the injected TRIDENTs gradually decreased with time in normal mouse, whilst it can retain till 48h in bacteria-infected mouse. To clarify the reason, H&E staining assay was used to characterize the structural changes of skin before and after bacterial infection. Different from the normal skin, the epidermis and dermis of the infected skin became thickened and denser, which would prevent nanoparticle permeation from the infected site, resulting in the long-time retention of TRIDENTs in infected site (Fig. 5b and Supplementary Fig. 10). Similar phenomenon has been reported in the literature^{6,7}. (Page 14, line 222-223; Page 16, line 253-254)

Supplementary Fig. 10. The retention of TRIDENTs after the local injection *in vivo*. a) Fluorescence images of skin obtained from healthy and infected mice. b) Corresponding histological photomicrographs of healthy and infected skin tissue sections. Scale bars: 20 μm .

11. It is suggested that the authors perform *in vivo* photothermal experiments for their developed TRIDENTs.

Response: Heating curve and thermal images of TRIDENT in infected skin under NIR irradiation had been added in the Supplementary Fig. 11. The temperature of the infected skin marked with white circles could be raised by 12 $^{\circ}\text{C}$. The final temperature could be maintained around 49 $^{\circ}\text{C}$, which would promote photothermal therapy and activate the injected TRIDENT to kill bacteria while reducing side effect generated from hyperthermia on surrounding normal tissues^{3,8,9}. (Page 14, line 225-229; Page 26, line 456-459).

Supplementary Fig. 11. *In vivo* photothermal experiments for the developed TRIDENT. a) Representative thermal images of mice obtained from the test process. b) Corresponding heating curve (n=3).

12. Fig. 6c. A typo of “imagines” as found.

Response: Thanks for your careful suggestion. This error has been corrected (as shown on Page 18, line 283).

13. Page 22, Line 11. The meaning of the sentence “the solution in one tube...for 24 h” is confusing.

Response: We have revised the text to make it clearer according to your suggestion as follows. “1 mL TRIDENT solution (500 $\mu\text{g}/\text{mL}$ for IMP) were separately added to four dialysis tubes (MWCO=3500D) and then were shaken in an orbital shaker. NIR irradiation was applied to the four tubes for 5, 10, 15, and 20 minutes in turn at the time point of 3 h, and then vibration again. At selected time intervals (0, 1, 2, 4, 6, 8, 10, 12 and 24 h), 500 μL released medium were retrieved and the absorbance at 294 nm was measured by High Performance Liquid Chromatography (HPLC) to determine the amount of the released IMP according to the calibration curve. The cumulative release was defined as: mass of released IMP/mass of loaded IMP \times 100%” (Page 23, line 386-392).

Reviewer #2:

This original work by Qing et al. proposes the use of drug and fluorophore loaded liposomes for use as a combinatorial therapy against bacterial infections. They have dubbed their system TRIDENT, and I applaud their efforts to make an acronym work, even if it's a little shaky. Laser application is used to heat the fluorophore, which disrupts the liposomal membrane, releasing an antibiotic in a local fashion. This method purports that the photothermal heating enhances bacterial killing and the local antibiotic release helps to reduce toxicity. On the whole, the work is very interesting, and shows promise. The use of the English language is acceptable, although some of the abbreviations probably need to be defined sooner (or at all). However, the authors omit several essential controls from their study that would elucidate the mechanisms at play. Additionally, there are several minor issues that need to be addressed. My concerns are listed below in no particular order. I recommend the authors revise the work and resubmit once essential changes have been made.

Thanks for your meaningful comments.

The essential controls that must be added to make this manuscript complete are:

A) Viability of bacteria in response to thermal stress akin to what is demonstrated with the TRIDENT system (e.g. +12 C for 5 min).

Response: Thanks for your advice. A control reflecting the viability of bacteria in response to thermal stress was added. Bacteria were exposed to various temperatures (37, 40, 43, 46, and 49 °C for 5 min) for different time (0, 1, 3, and 5 min at 49 °C) to test their thermal response and the results were showed in Supplementary Fig. 6 and 7. Along with the temperature increased rose to 46 °C, the viabilities of the four kinds of bacteria were decreased by about 20%, while nearly half of the bacteria were inactive after the treatment at 49 °C for 5 min. Similar results were obtained when the bacteria were treated at 49 °C for different times, indicating that bacteria can be partially killed at the relatively high temperature¹⁰. (Page 11, line 166-173; Page 25, line 419-427)

Supplementary Fig. 6. Thermal response of four kinds of bacteria to various temperatures. a) Images of the colonies formed on LB-agar plates. b) Statistical analysis of the bacterial cell viability.

B) There needs to be an IMP treatment group based on the theoretical released IMP dose rather than 3x the MIC, which is a relatively arbitrary value.

Response: An IMP group based on the theoretically released IMP was added as requested (Fig. 3 and 4, Supplementary Fig. 12). In our study, the triple IMP group works as a positive control to verify whether antibiotics are able to kill all bacteria at high doses. We found that antibacterial activity of 3x IMP was inferior to TRIDENT+NIR (Page 11, line 161-166; Page 13, line 201-203).

C) All thermal treatments are TRIDENT, there needs to be a control thermal group that uses the TRIDENT platform without IMP.

Response: We are sorry about the confusion caused by the abbreviation in our previous manuscript. We have replaced the G1-G7 with full name for each group in the revision. And the group of IR780@TRN+NIR reflecting the control thermal group for TRIDENT+NIR (namely IMP/IR780@TRN+NIR) has been included already.

1) Since the G1-G7 groups are referenced heavily in the Figure and the text, a table describing these groups would be extremely helpful.

Response: According to your suggestion, the groups of G1-G7 appearing in the manuscript have been replaced with full name in the revision (Page 11, line 162-163; Page 11, 173-180; Page 12, line 195; Page 15, line 235-238; Page 15, line 242-246; Page 22, line 368-372; Page 24, line 405-407; Page 27, line 461-466).

2) The hydrodynamic and core diameters need to be more precisely defined that "~40 nm" particularly if the size distribution is "within 50 - 80 nm." Specific values and standard deviations (or SEMs) are essential. This is particularly important since later remarks are made regarding the percentage of particles in the 80-100 nm size range.

Response: We have corrected these contents and the standard deviations were expressed throughout the text. We have corrected the diameters with specific values and standard deviation (63.39 ± 7.75 nm with a PDI of 0.27 ± 0.05) in the text (Page 6, line 96-98). And the data have been updated in Fig. 1 (Page 6, line 109).

3) Similarly standard deviations are needed for the loading and encapsulation efficiency. It would also be helpful for the authors to define these terms precisely in the text where relevant. I would also caution the authors to pay attention to significant Figure when standard deviations are added. In general, the authors need to include precise numbers throughout the results section.

Response: Yes. Thanks for your advice. The standard deviations have been added throughout the manuscript (Page 6, line 101). And two significant figures with precise numbers were kept when the standard deviations were added throughout the results section.

4) Fig. 5 indicates that the fluorescence is the result of tail vein injections (implying some form of selective localization at the site of infection), which is NOT described in the methods (subcutaneous) or main text. If this is an error it needs to be corrected.

Response: Thanks for your comments. Actually we injected the nanoparticle via subcutaneous injection instead of tail vein injections. The blurry dots at the mouse tail are the labels we made to differentiate the mice (Page 16, line 254).

5) At least one citation should be included with the claim that 12 C "could lead to serious damage to cell walls". Expanding on this, it would be extremely beneficial for the authors to show the response of these bacteria to various temperatures and times that are relevant to the study.

Response: The reference supporting the claim have been cited in the manuscript¹¹ (Page 8, line 126-127). We investigated the bacterial viability by standard plate count and found that the survival rate of bacteria was negatively proportional to the extent of treatment temperature and time. For a certain time, the increased temperature will lead to less survival rate once the temperature rises to 46 °C and above (Supplementary Fig. 6,7). (Page 11, line 166-173; Page 25, line 419-427)

Supplementary Fig. 6. Thermal response of four kinds of bacteria to various temperatures. a) Images of the colonies formed on LB-agar plates. b) Statistical analysis of the bacterial cell viability.

6) The inclusion of the DSC data is a welcome change from most liposome studies. However, it would be extremely valuable to include data from drug and/or dye loaded TRIDENT to confirm that the thermal properties are not changed as a result of these new contributions.

Response: We have added the DSC data in different status of TRIDENT (IMP and/or IR780 loaded) in the revised manuscript (Fig. 2a) (Page 9, line 130-131). And the phase-change temperature of the IMP and/or IR780 loaded TRIDENT did not show obvious change compared to the carrier that composes of lauric acid and stearic acid.

Fig. 2a. DSC curves of TRN, IMP@TRN, IMP/IR780@TRN, lauric acid (LA) and stearic acid (SA).

- 7) The introduction should provide a more thorough literature review regarding combinatorial PTT and antibiotic therapy. This is an emerging area of research, and the various approaches should be addressed. In particular, use of other nanomaterials, such as graphene, gold, and silver should be addressed given their inherent photothermal capacity.

Response: The photothermal capacity of other nanomaterials has been added in the section of introduction (Page 3-4, line 52-62) as follows. “The use of varied inorganic or polymer nanoparticles for bacterial killing based on their inherent photothermal capacity has been intensively reported. Besides, nanoparticles such as AuNPs, iron oxide, graphene, black phosphorus and other polymer nanoparticles capable of photothermal conversion have been designed to intelligently deliver antibiotics to the infected sites for chem-photothermal therapy under NIR irradiation^{8, 9, 12, 13}. These synergetic strategies can give full play to the advantages of both and enhance the antibacterial effect as well as reduce the side effects under the control of external NIR irradiation. IR-780 iodide (IR780) is a near-infrared (NIR) fluorescence dye with high and stable fluorescence intensity and have been utilized in photothermal therapy¹⁴. Different from the inorganic nanomaterials, IR780 has better degradability and lower long-term toxicity, which make it more suitable for *in vivo* photothermal therapy. Besides, the inherent characteristic of the lipophilicity makes it easy to be wrapped by hydrophobic carriers¹⁵.”

- 8) The bacterial viability data should be presented in a log scale rather than a percent. It is not instructive in a situation where $1e6$ cells are initially treated to show that <1% CFUs remain. There can be literally thousands of cells remaining and an additional day of culturing may show that valuable therapeutic effect has been realized.

Response: We quite agree the suggestion and have corrected the viability in a log scale both in Fig. 3 and 4 (Page 11, line 175-180; Page 12, line 189-190; Page 14, line 212-213). In the previous experiment, we only counted the viability according to the counted CFU in each petri dish and forgot to calculate the dilution folds. In the revision, we have corrected the formula and expressed in a log scale. These corrected contents are shown on Page 24, line 410-415. “The groups of IR780@TRN+NIR, IMP/IR780@TRN+NIR and NIR were separately irradiated with an 808 nm laser at a power density of 0.5 W cm^{-2} for 5 min. Then, the solution (50 μL) of the bacterial suspension was spread on LB plates followed by 100x dilution of the solution, and the number of colony-forming units (CFU) in each plate was counted and calculated by the form of $\log_{10}\text{CFU/mL}$ to decide the antibacterial effect of different conditions after cultivation for 24 h at 37 °C (n=3).”

- 9) Is IR780 being used as a fluorescent tracker/photothermal transducer or is there an additional purpose as a photosensitizer?

Response: Yes. No purpose other than fluorescent tracker and photothermal transducer for the IR780 in this study.

- 10) Does colloidal stability simply mean the hydrodynamic diameter did not change? More importantly, are there other changes, like leakage, that occur during the process?

Response: The colloidal stability evaluation of the TRIDENTs is characterized by the hydrodynamic diameter change. We monitored the HD evolution of the TRIDENTs incubated in various solvents for 7 days and no significant changes on HD of TRIDENTs were found. In addition, the morphology of the TRIDENT was also tested by TEM and no obvious change was obtained in the image, which implies no obvious leakage of the TRIDENT. And we have corrected the inappropriate statement of “colloidal stability” to “size and morphology” to make it more accurate in the main text (Page 6, line

102) and Supplementary Fig. 2.

Supplementary Fig. 2. TEM images of TRIDENTs in four solvents (H₂O, 0.9% NaCl (NS), PBS and Luria–Bertani broth medium (LB)) that obtained in 1st and 7st day. Scale bars: 200 nm.

11) The release profile is unimpressive. In 24 hours roughly 15% of the loaded drug leaks out, while laser treatment increases the release from 15% to a whopping 30%. Why is this thermally triggered release so meager? Where is the rest of the drug? By extension, the schematic is misleading as it suggests that the liposome will be fundamentally destroyed at the elevated temperature (indicating release on the order of 100%). So it is probably wise to revise the schematic as well. The authors do comments that the TRNs are partially damaged from the photothermal treatment, but the extent of this damage and the underlying implications are left unstated. Additionally, the loading of IMP seems extremely low.

Response: We repeated the IMP release test of TRIDENT and replaced the release profile curve according to the comment. To achieve more accurate drug release data, HPLC was used to determine the released IMP. It was observed that increased irradiation time induced longer temperature maintenance, this induced an elevated IMP release (Fig. 2f) (Page 9, line 134-135). Whilst the melted TRN trended to solidify, and the partially released IMP will be encapsulated by TRN again when the temperature decreased below 43 °C. Then, non-encapsulated IMP would slowly diffuse into the medium afterward and cumulative release could be detected finally³. Our latest TEM images demonstrated that the TRIDENTs became smaller after NIR irradiation, revealing substantial damage of TRIDENTs and nearly 80% cumulative IMP release could be observed at 20 min NIR irradiation (Fig. 2f) (Page 10, line 144-151; Page 23, line 386-392). The TRIDENT loaded with more drugs has been prepared through the improved method.

Fig. 2f. *In vitro* cumulative release profile of IMP from TRIDENT under NIR irradiation with different time.

12) Fig4e shows similar membrane damage in quite a few panels more than is discussed in the text.

Response: We replaced the SEM images with clearer ones, from which different membrane damage could be observed between the images of TRIDENT+NIR and IR780 TRN+ NIR group (Fig. 3f,4f and Supplementary Fig. 9). More conspicuous membrane damage could be observed in TRIDENT+NIR group than IR780@TRN+NIR group. (Page 12, line 192; Page 14, line 215)

Supplementary Fig. 9. SEM images of MDREC and EC. Scale bars: 1 μm .

13) I appreciate the addressing of biosafety, but the authors should make clear that this is an acute rather than chronic study.

Response: Yes. Acute biosafety was addressed in the revised manuscript (Page 17, line 272).

14) It is inadequate to use a methanol calibration in an aqueous system without justifying their equivalence.

Response: Thanks for your meaningful advice. We have justified the methanol equivalence to remove the effect of methanol by using the following experiment. Briefly, 10 μL IMP or IR780 with a serial of concentrations and 990 μL methanol were exploited for primary calibration. Then, 10 μL sample with unknown concentration was added into 990 μL methanol for UV-vis test, by which the error from methanol could be eliminated. (Page 22-23, line 372-374)

15) The discussion is more or less useless. It's a summary of the findings that provides no real insight into the platform nor does it discuss the limitations or potential improvements.

Response: We revised the discussion section of the TRIDENT and limitations as well as potential improvements were also proposed (See Page 19, line 296-306; Page 19, line 307-309; Page 20, line 332-338).

Discussion:

line 296-306:

Secondly, NIR-irradiated IR780 molecules will generate a thermal response that not only melts the TRN but also damage the cell membrane via photothermal effect, which will assist with the third function for reducing the resistance to IMP and promoting the rapid diffusion of IMP into cytoplasm to chemically interfere with bacterial cell wall synthesis. Both antibiotic-sensitive and clinical MDR bacteria can be killed *in vitro* (Fig. 3,4) and *in vivo* (Fig. 5, Supplementary Fig. 12,13,14) via the highly effective chemo-photothermal therapy. The highly effective combinational antimicrobial activity may help to avoid the further production of drug-resistant bacteria. Additionally, by encapsulation within TRN, the degradation and nephrotoxicity of IMP by renal enzyme dehydropeptidase 1, usually avoid through co-administered with cilastatin in clinic¹⁶, will be sharply reduced when used *in vivo*, which will reduce the dosage of IMP and enhance the biosafety

(Supplementary Fig. 14)^{15,17}.

Lines 307-309:

It is notable that NIR-irradiated TRIDENT killed MRSA, frequently resistant to β -lactam antibiotics including IMP, with an enhanced synergistic effect compared to antibiotics or photothermal therapy alone (Fig. 3,4,5, and Supplementary Fig. 12,13,14).

Lines 332-338:

Some defects or limitations are still existed in this study. Firstly, the drug-loading efficiency could be further improved although they are comparable to some existing carriers. Secondly, more accurate antibacterial mechanisms are needed to be further elucidated. Thirdly, the strategy is inefficient in treating deeper tissue infections or systematic blood infections owing to the limited tissue penetration capacity and irradiation area of NIR light^{6,18,19}. Therefore, developing strategies employing other photosensitive molecules excited by NIR-II region light and encapsulating additional functional molecules could further improve the potential clinic application of the as proposed TRIDENT strategy.

Reviewer #3:

This manuscript describes the phase changeable fatty acid-based nanotransporter for bacteria therapy. The system consisted of fatty acid mixture of lauric acid (LA) and stearic acid (SA) at 3.5:1 in weight, that can be melted in range of 43 ~ 44 °C. The mixture was used as a gate material for on-demand release of pre-loaded antibacterial agent without undesired passive diffusion of the agent. Addition of IR780, a photothermal agent capable of converting light energy into heat, mediated bacterial lysis by denaturing the bacterial membrane proteins. Consequently, the chemo-photothermal effect could successfully kill the bacteria. Although the synergistic therapy using phase changeable fatty-acid based nanoparticles has been already reported by Younan Xia and coworkers, especially for cancer therapy, this work is very impressive in that the use of fatty acid-based nanoparticles was achieved for the treatment of bacterial infection. Besides, the biosafety of the nanoparticles was investigated using in-vivo mouse model, which has not been reported, to the best of my knowledge, in previous studies. The experimental design and data of this study are clear to convince to the conclusion. I strongly support the publication of this work after several things are clarified.

1. According to previous literatures studying on the mixture of LA and SA (J. Chem. Eng. Data 2011, 56, 2889; Macromol. Mater. Eng. 2016, 301, 887), the mixture has only one melting point (called eutectic point) at a specific composition. In the other forms, the mixture has another melting DSC peak as well as the eutectic peak. However, in this work, each composition of LA and SA had one melting point, as shown in Fig. S2. The authors need to explain this result more detail.

Response: We repeated the DSC test of the composition of LA and SA with a slow heated rate (5 °C per min) and the eutectic peak along with another melting peak can be observed (Supplementary Fig. 3). In our previous experiment, the heated rate (10 °C per min) may be too fast that missed the signal of eutectic peak.

Supplementary Fig. 3. DSC measurements for TRN with different mass ratios of LA/SA.

2. The authors irradiated NIR light to TRIDENT for studying on the morphology of the system and its anti-bacteria ability. When the light was exposed, what was the temperature of the system?

Response: The initial temperature of the system was 37°C before laser exposure. And the final temperature was raised to 49 °C at the end of irradiation (0.5 W/cm², 5 min). The temperature change was monitored with an infrared thermal camera in a real-time manner.

Response Fig. 1. Thermal images of TRIDENT solution under NIR irradiation (0.5 W cm^{-2}) before and after NIR irradiation.

Response Fig. 2. Heating curves of the test solution and the thermal images of the solution at start and end point were inset in the figure.

- The authors stated the photothermal effect of IR780 in response to NIR light melted TRN. But, as shown in Fig. 2d and 2e, there was no significant change in the morphology and size of the system and the diminutive particles were formed. If the melting completely occurred, I think the formation of the much more diminutive particles should be observed due to the destruction of the fatty-acid based nanoparticles, as illustrated in Fig. 2g. Why was the morphology and size of the TRN unchanged after NIR irradiation? Was it attributable to non-uniform distribution of IR780 within TRN, leading to partial damage?

Response: In our previous test, the samples for TEM observation and DLS measurements were derived from the TRIDENT solution, which was naturally cooled down to room temperature after laser irradiation. However, the TRN could naturally solidify into nanoparticles during the slowly cooling down process because of the intrinsic hydrophobicity of the inner core (lauric acid and stearic acid). Therefore, it is challenging for TEM observation to veritably reflect the morphology of TRIDENT immediately at the end of irradiation. Of note, to the best of our knowledge, the published literatures have barely provided and employed convinced TEM images to validate the morphology change raised from laser irradiation to date¹⁻³. Maybe it is the above proposed dilemma that hinders researchers to use TEM observation for the study.

To realize more accurate investigation on the morphology change of TRIDENT using TEM, the TRIDENT solution was immediately immersed in liquid nitrogen for rapid solidification at the end of laser irradiation. Due to the prompt decrease of the temperature, fractured TRIDENTs would immediately solidify and the instantaneous morphology of them could be held without any other changes. After cooling down in liquid nitrogen for 1 min, the solidified solution was naturally warmed up to room temperature ($\sim 25 \text{ }^\circ\text{C}$, lower than the phase-transition temperature of the inner core).

Thereby, the irradiated TRIDENTs retained their original morphology at the end of laser irradiation and suited for TEM observation (Page 23, line 380-384).

The temperature of the TRIDENTs solution was increased from $\sim 37\text{ }^{\circ}\text{C}$ to $\sim 49\text{ }^{\circ}\text{C}$ upon the laser irradiation (Response Fig. 1). Meanwhile, compared with the TRIDENTs before irradiation, many smaller nanoparticles were observed by TEM from the sample of TRIDENTs upon laser irradiation. Taking together with the decrease of hydrodynamic diameters, the morphology change suggested that the nanostructure of TRIDENTs was destroyed during the process of laser irradiation and thus thermal-responsive (Fig. 2d). Similar phenomenon was also found in IR780@TRN (Supplementary Fig. 4). We have revised the relative content in the manuscript (Page 10, line 138-142).

Supplementary Fig. 4. TEM images and particle size distribution histograms of IMP/IR780@TRN, IR780@TRN and IMP@TRN before and after the NIR irradiation. Scale bar: 200 nm.

4. When the release of IMP from TRIDENT occurs under NIR light, it seems that IR780 can be released as well. How much amount of the photothermal agent was released?

Response: Yes. It is correct that both IMP and IR780 could be released under NIR light irradiation. We found $23.13\pm 0.32\%$ of total IR780 (equivalent to $69.39\pm 0.96\text{ }\mu\text{g}$) was released after NIR irradiation for 5 min (Response Fig. 4).

Response Fig. 4. Cumulative release profile of IR780 from TRIDENT under NIR irradiation for 5 min *in vitro*.

5. In bacterial test, the treatment with only NIR irradiation need to be demonstrated in order to clarify the anti-bacterial ability of TRIDENT + NIR.

Response: We added a group with NIR irradiation alone in both *in vitro* and *in vivo* antibacterial experiments (Fig. 3 and 4). The recovery degree of the infected mice skin under NIR irradiation alone was similar with that treated with PBS, indicating that NIR irradiation alone produces negligible therapeutic effect (Supplementary Fig. 12).

Supplementary Fig. 12. Antibacterial activities of NIR irradiation, T-IMP, 3x IMP and TRIDENT for the MRSA-infected skin *in vivo*. a) Representative photographs of the infected skin of mice during treatment with different formulations within 15 days. b) Size of the infected area and c) body weights of the mice. d) Histological photomicrographs of skin tissue sections of infected mice after completion of the antibacterial activity experiment. Data are presented as mean ± S.D. (n=3).

Minor Issues.

1. Fig. 3e, 4e shows SEM images not TEM images.

Response: Yes. We have corrected this in the revision (Fig. 3e, 4e). (Page 12, line 192; Page 14, line 215).

References

1. Zhu, C. et al. A eutectic mixture of natural fatty acids can serve as the gating material for Near-Infrared-triggered drug release. *Adv. Mater.* **29** (2017).
2. Liu, G. et al. “Wax-Sealed” Theranostic nanoplatform for enhanced afterglow imaging–guided photothermally triggered photodynamic therapy. *Adv. Funct. Mater.* **28**, 1804317 (2018).
3. Zhao, Y. et al. Near-Infrared light-activated thermosensitive liposomes as efficient agents for photothermal and antibiotic synergistic therapy of bacterial biofilm. *ACS Appl. Mater. Inter.* **10**, 14426-14437 (2018).
4. Lee, A.S. et al. Methicillin-resistant *Staphylococcus aureus*. *Nat. Rev. Dis. Primers* **4**, 18033 (2018).
5. Tan, C.M. et al. Restoring methicillin-resistant *Staphylococcus aureus* susceptibility to β -lactam antibiotics. *Sci. Transl. Med.* **4**, 126ra135-126ra135 (2012).
6. Korupalli, C. et al. Acidity-triggered charge-convertible nanoparticles that can cause bacterium-specific aggregation in situ to enhance photothermal ablation of focal infection. *Biomaterials* **116**, 1-9 (2017).
7. Wang, F. et al. Hydrogel retaining toxin-absorbing nanosponges for local treatment of methicillin-resistant *Staphylococcus aureus* infection. *Adv. Mater.* (2015).
8. Ji, H. et al. Bacterial hyaluronidase self-triggered prodrug release for chemo-photothermal synergistic treatment of bacterial infection. *Small* **12**, 6200-6206 (2016).
9. Meeker, D.G. et al. Synergistic photothermal and antibiotic killing of biofilm-associated *Staphylococcus aureus* using targeted antibiotic-loaded gold nanoconstructs. *ACS Infect. Dis.* **2**, 241-250 (2016).
10. Wu, M.-C., Deokar, A.R., Liao, J.-H., Shih, P.-Y. & Ling, Y.-C. Graphene-based photothermal agent for rapid and effective killing of bacteria. *ACS Nano* **7**, 1281-1290 (2013).
11. Wu, M.C., Deokar, A.R., Liao, J.H., Shih, P.Y. & Ling, Y.C. Graphene-based photothermal agent for rapid and effective killing of bacteria. *ACS Nano* **7**, 1281-1290 (2013).
12. Chiang, W.-L. et al. A rapid drug release system with a NIR light-activated molecular switch for dual-modality photothermal/antibiotic treatments of subcutaneous abscesses. *J. Controlled Release* **199**, 53-62 (2015).
13. Hu, B., Zhang, L.-P., Chen, X.-W. & Wang, J.-H. Gold nanorod-covered kanamycin-loaded hollow SiO₂ (HSKAurod) nanocapsules for drug delivery and photothermal therapy on bacteria. *Nanoscale* **5**, 246-252 (2013).
14. Kuang, Y. et al. Hydrophobic IR-780 dye encapsulated in cRGD-conjugated solid lipid nanoparticles for NIR imaging-guided photothermal therapy. *ACS Appl. Mater. Inter.* **9**, 12217-12226 (2017).
15. Yue, C. et al. IR-780 dye loaded tumor targeting theranostic nanoparticles for NIR imaging and photothermal therapy. *Biomaterials* **34**, 6853-6861 (2013).
16. Tahri, A. et al. Exposure to imipenem/cilastatin causes nephrotoxicity and even urolithiasis in Wistar rats. *Toxicology* **404-405**, 59-67 (2018).
17. Li, D. et al. Tumor acidity/NIR controlled interaction of transformable nanoparticle with biological systems for cancer therapy. *Nano Lett.* **17**, 2871-2878 (2017).
18. Yang, K. et al. The influence of surface chemistry and size of nanoscale graphene oxide on photothermal therapy of cancer using ultra-low laser power. *Biomaterials* **33**, 2206-2214 (2012).
19. Hsiao, C.-W. et al. Effective photothermal killing of pathogenic bacteria by using spatially tunable colloidal gels with nano-localized heating sources. *Adv. Funct. Mater.* **25**, 721-728 (2015).

REVIEWERS' COMMENTS:

Reviewer #1 (Remarks to the Author):

The authors have significantly improved the quality of the manuscript. They have carefully addressed all the suggestions and concerns that had been raised by this reviewer with detailed explanation and additional new data. Particularly, they have elucidated the novelty of the concept/design of their study. Therefore, this manuscript is acceptable for publication in Nature Communications in its current form.

Reviewer #2 (Remarks to the Author):

I thank the authors for meticulously addressing my concerns. I recommend the authors address several cosmetic issues before the paper is accepted/published.

Insets in Figure 2D & E are difficult to read because the background SEM.

The text in figure 3 a, d, and e is hard to read due to the font size. While the X-axis in 3b and c are almost impossible to discern. The same can be said of figure 4. If necessary, some of these components can be sent to the SI. I have similar concerns with figure 5 c-k

Reviewer #3 (Remarks to the Author):

The authors have revised the ms very well with the elaborate consideration regarding the suggested issues.